# Improving a probabilistic cytoarchitectonic atlas of auditory cortex using a novel method for inter-individual alignment

Omer Faruk Gulban[1,2]*, Rainer Goebel[1,2], Michelle Moerel[1,3], Daniel Zachlod[4,5], Hartmut Mohlberg[4,5], Katrin Amunts[4,5], Federico de Martino[1,6]

[1]Department of Cognitive Neuroscience, Maastricht University, Maastricht, Netherlands; [2]Brain Innovation B.V, Maastricht, Netherlands; [3]Maastricht Centre for Systems Biology, Faculty of Science and Engineering, Maastricht University, Maastricht, Netherlands; [4]Institute for Neuroscience and Medicine (INM-1), and JARA Brain, Research Centre Jülich, Jülich, Germany; [5]C. and O. Vogt Institute for Brain Research, Heinrich Heine University, Düsseldorf, Germany; [6]Center for Magnetic Resonance Research, University of Minnesota, Minneapolis, United States

**Abstract** The human superior temporal plane, the site of the auditory cortex, displays high inter-individual macro-anatomical variation. This questions the validity of curvature-based alignment (CBA) methods for in vivo imaging data. Here, we have addressed this issue by developing CBA+, which is a cortical surface registration method that uses prior macro-anatomical knowledge. We validate this method by using cytoarchitectonic areas on 10 individual brains (which we make publicly available). Compared to volumetric and standard surface registration, CBA+ results in a more accurate cytoarchitectonic auditory atlas. The improved correspondence of micro-anatomy following the improved alignment of macro-anatomy validates the superiority of CBA+ compared to CBA. In addition, we use CBA+ to align in vivo and postmortem data. This allows projection of functional and anatomical information collected in vivo onto the cytoarchitectonic areas, which has the potential to contribute to the ongoing debate on the parcellation of the human auditory cortex.

**\*For correspondence:**
faruk.gulban@
maastrichtuniversity.nl

**Competing interests:** The authors declare that no competing interests exist.

## Introduction

Historically, there has been a substantial effort to describe the micro-anatomy of the human auditory cortex (*Von Economo and Horn, 1930*; *Galaburda and Sanides, 1980*; *Rivier and Clarke, 1997*; *Morosan et al., 2001*; *Wallace et al., 2002*; *Morosan et al., 2005*; *Clarke and Morosan, 2012*; *Nieuwenhuys, 2013*). Various parcellation schemes have been proposed, which identify a primary area (core; primary auditory cortex) as well as secondary belt and tertiary parabelt auditory areas (*Rivier and Clarke, 1997*; *Moerel et al., 2014*). The primary auditory cortex (PAC) is generally located on the medial two-thirds of Heschl's Gyrus.

It has proven challenging to use these results to identify auditory areas in individuals in vivo, as classical cyto- (and myelo-) architectural approaches are limited by the absence of an objective metric defining cytoarchitectonic areas. In addition, relating micro-anatomical characteristics to macro-anatomy is hampered by the inherent two-dimensional representation of the results (i.e. by means of drawings or labeled slices) and scarce information regarding inter-subject variability. Instead, observer-independent methods for the analysis of serial cytoarchitectonically stained sections, that additionally correct for shrinkage artifacts typical of histological processing (*Amunts et al., 2000*), have been developed in the last 20 years (*Schleicher et al., 1999*). Using this method,

*Morosan et al., 2001* identified various auditory areas in the superior temporal cortex and generated a probabilistic atlas based on 10 individual brains. This atlas (*Eickhoff et al., 2005*) allows assigning probabilistic values to in vivo brain images and has been used to, for example, validate the delineation of PAC on the basis of in vivo MRI images whose contrast is related to myelin (*Dick et al., 2012*).

The probabilistic atlas is generated using a volume registration method. Instead, the exceptionally reliable correspondence between micro- and macro-anatomy known to be present in many cortical areas (*Turner, 2013*) has inspired the use of registration methods that rely on cortical surfaces and macro-anatomical landmarks such as the major gyri and sulci (i.e. curvature-based alignment [CBA] rather than the whole volumetric data [*Fischl et al., 1999*; *Frost and Goebel, 2012*; *Goebel et al., 2006*]). Surface-based alignment methods have been shown to improve the accuracy of inter-individual registration in micro-anatomically defined primary motor cortex (*Fischl, 2013*), the human middle temporal area (hMT) (*Frost and Goebel, 2013*), and to improve the registration of a cytoarchitectonic atlas of the ventral visual system (*Rosenke et al., 2018*; *Fischl et al., 2008*).

With the aim of minimizing inter-individual variability and thus improving (statistical) power when mapping functional properties, CBA is also routinely used in studies investigating the functional and anatomical properties of auditory cortical areas. Beyond the use of whole brain curvature patterns, local landmark alignment approaches have also been used for aligning temporal cortical regions (*Kang et al., 2004*; *Desai et al., 2005*). These methods have been shown to outperform approaches that optimize whole brain features on the basis of both their ability to obtain clearer gyral and sulcal structures as well as higher resolution functional maps at the group level. However, Heschl's Gyrus substantially varies in shape across individuals and across hemispheres, and slight changes in the primary auditory cortex location have been reported in subjects with a typical morphological variation of the Heschl's Gyrus (*Heschl, 1878*; *Rademacher et al., 1993*; *Hackett et al., 2001*; *Marie et al., 2015*. Given this variation in superior temporal plane macro-anatomy across individuals and shift of micro-anatomical areas with macro-anatomy, it is debatable if curvature-based alignment (either based on whole brain or local landmarks) improves the correspondence of micro-anatomically defined auditory areas and their corresponding functional characteristics.

To address this issue, here we applied curvature based alignment (abbreviated as CBA), as well as a procedure tailored to the temporal lobe by incorporating anatomical priors (abbreviated as CBA+). By evaluating the inter-individual alignment resulting from these procedures (and comparing them to the original volumetric approach), we aimed at justifying the use of CBA or CBA+ in functional studies of the temporal lobe. In particular, we reconstructed cortical surfaces from the data of *Morosan et al., 2001* and investigated the effect that maximizing macro-anatomical inter-individual alignment has on the overlap of micro-anatomically defined auditory cortical areas. We reasoned that a method that provides a more accurate inter-individual alignment of (postmortem) cytoarchitecture will consequently provide a more accurate alignment of the associated functional properties. Thereby, here we validate the use of CBA in previous studies and demonstrate that CBA+ is a better tool for aligning the superior temporal plane across participants. Moreover, we provide an improved probabilistic surface atlas of auditory cortical regions as a publicly available resource for the auditory community (*Gulban, 2020*; copy archived at https://github.com/elifesciences-publications/cortical-auditory-atlas). We showcase this approach by applying CBA+ to an in vivo dataset collected at 7 Tesla and projecting the improved cytoarchitectonic atlas onto functional and anatomical group maps. In addition, in order to contribute to the ongoing debate on the in vivo localization of auditory cortical areas (*Moerel et al., 2014*; *Besle et al., 2018*), we align the cytoarchitectonic atlas (and in vivo data) to a recent temporal lobe parcellation based on in vivo measurements (*Glasser et al., 2016*).

## Results

We obtained cytoarchitectonically labeled temporal cortical areas and postmortem MR images of 10 brains (volumetrically aligned (rigid body) to the Colin27 space) used in the JuBrain cytoarchitectonic Atlas (*Amunts and Zilles, 2015*). The cytoarchitectonically labeled areas were TE 1.0, TE 1.1, and TE 1.2 from *Morosan et al., 2001*, TE 2.1 and TE 2.2 from *Clarke and Morosan, 2012*, TE three from *Morosan et al., 2005*, and STS one and STS two from *Zachlod et al., 2020*. In order to perform cortex based alignment, the white matter - gray matter boundary was segmented in all 10 postmortem

brains. To obtain this segmentation, we have used a combination of image filtering techniques and a histogram-based segmentation approach (*Gulban et al., 2018b*), which reduced the amount of required manual corrections (see Materials and methods section). Cortical surfaces were reconstructed to perform three different types of group alignment methods. These methods were rigid body (i.e. considering surface sampling [compared to volumentric alignment] and rigid body registration), CBA and CBA with anatomical priors (CBA+; including the anterior Heschl's Gyrus, the superior temporal gyrus, the superior temporal sulcus, and the middle temporal gyrus as anatomical priors). We additionally compared these surface approaches to the original volumetric alignment in the Colin27 space. We have validated the performance of these methods by comparing the overlap between cytoarchitectonic areas across individuals. We subsequently used CBA+ to create superior temporal cortical group maps of in vivo MRI (at 7T) measurements and to align them to the probabilistic cytoarchitectonic atlas.

## Comparison between alignment methods

*Figure 1* rows 1 and 3 show the averaged curvature maps after alignment with each of the surface approaches we used (i.e. rigid only that linearly coregisters the surfaces, standard CBA, and CBA tailored to the temporal lobe [CBA+]). In the temporal lobe, the increased sharpness of the average curvature maps indicates the improved correspondence of the macro-anatomical features in CBA and CBA+ compared to the rigid only alignment. Especially in the right hemisphere (third row in *Figure 1*), an improvement of CBA+ over standard CBA is noticeable at the level of the Heschl's Gyrus (indicated by a red circle). The improvement in alignment of the macro-anatomical features in the temporal lobes (left and right) is also visible when considering the folded average meshes of the ten brains in the postmortem dataset (i.e. average folded meshes, *Figure 1* rows 2 and 4). In absence of large macro-anatomical differences across the individuals, improved alignment should increase the 3D complexity (e.g. gyri and sulci appearing very clearly distinguishable) of the average folded mesh. Cortical curvature-based alignment procedures, however, may be affected when individual cortical macro-anatomy strongly deviates from the average morphology. In the postmortem sample, we analyzed, we observed macro-anatomical variations across hemispheres of two types. First, following the characterization described in *Kim et al., 2000*; *Da Costa et al., 2011*, the number of Heschl's Gyri varied. In particular, we observed 1, 1.5 and 2 Heschl's Gyri in [5, 4, and 1, respectively] right hemispheres and [6, 2, and 2, respectively] left hemispheres. Second, we observed the presence of three hemispheres (one right and two left ones) whose single Heschl's Gyrus was continuous at the anterior part of the anterior temporal convolution, resulting in a split superior temporal gyrus (i.e. interrupted by an intermediate sulcus between the anterior and posterior part with respect to the location of the Heschl's Gyrus - Figure 13 lower right panel). This rare morphological pattern was first described in *Heschl, 1878* and was reported to occur % 10 of all brains inspected by Richard L. Heschl (110 of 1087 brains). It was eight times more likely to occur on the left hemisphere in comparison to right (also see *Rademacher et al., 1993*), for another reference to Heschl's work in English). As expected, the tailored alignment we developed here results in a more prominently defined Heschl's Gyrus in the average mesh, resulting from the correct alignment of the anterior Heschl's Gyrus across individual hemispheres. In the split superior temporal gyrus cases, we defined the gyrus as continuous (i.e. bridging the intermediate sulcus). While this definition did not compromise the alignment of the anterior Heschl's Gyrus, the impact of the approach we followed in the alignment of regions in proximity to the intermediate sulcus would require a larger sample on which to evaluate alignment separately according to this macro-anatomical variation (i.e. aligning separately individuals with a split/continuous superior temporal gyrus).

To evaluate the effect that minimizing macro-anatomical differences (as evidenced by the improved average curvature maps and folded meshes) has on micro-anatomy, we considered the inter-individual overlap of the cytoarchitectonically defined areas. In *Figures 2*, *3*, *4*, *5*, *6*, *7*, we present (for each labeled area) probabilistic maps (after alignment) indicating the number of subjects for which a given vertex is labeled as belonging to the same cytoarchitectonic area. For all cytoarchitectonic areas, CBA+ improves the overlap (as indicated by the increased probability of a vertex to be labeled as belonging to same area across the ten brains).

To better understand the differences between methods and quantitatively compare the rigid alignment, CBA, and CBA+ surface approaches to the initial volumetric alignment (in Colin27 space), *Figure 8* and *Figure 9* present the histograms of the probabilistic maps of each area (left and right

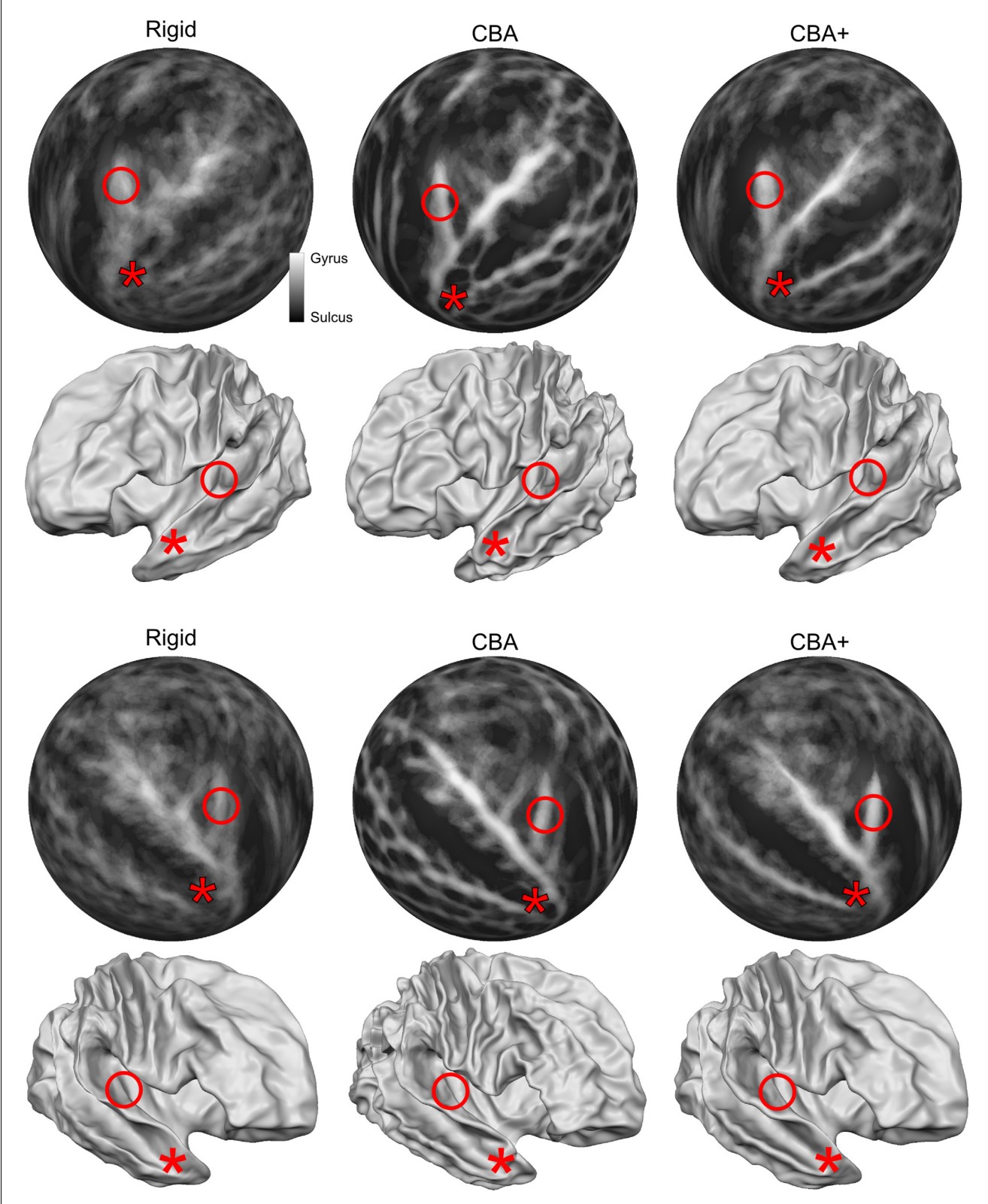

**Figure 1.** Differences between spherical rigid body alignment, curvature-based alignment (CBA), and CBA with an anatomical prior (CBA+) on group average binarized curvature maps visualized as half-sphere projections (rows 1 and 3) and group average vertex coordinates visualized as folded surfaces (rows 2 and 4). In rows 1 and 3, higher contrast between sulci (dark gray) and gyri (light gray) shows more overlap around Heschl's Gyrus which

*Figure 1 continued on next page*

indicates that a method better accounts for inter-subject morphological variation. In rows 2 and 4, the average vertex coordinates show a more pronounced Heschl's Gyrus in 3D as the alignment method improves the anterior Heschl's Gyrus overlap.

hemisphere, respectively). In addition, *Table 1* and *Table 2* report the dice coefficient obtained by building an atlas using 9 out of 10 of the hemispheres and computing its overlap with the areas of the left out hemisphere (leave-one-out procedure; left hemispheres in *Table 1* and right hemispheres in *Table 2*). Both the histograms and the dice coefficients indicate that, for the cytoarchitectonic areas along Heschl's Gyrus (Te1.0, Te1.1 and Te1.2), the largest overlap is provided by CBA+, which improves micro-anatomical correspondence compared to the volume-based alignment and the two other surface approaches we evaluated. For the areas in the planum temporale (Te2.1 and Te2.2), all surface approaches improve micro-anatomical correspondence compared to the volume alignment, and CBA+ provides an additional benefit especially for the area Te2.1. Similarly, for the areas in the superior temporal gyrus and sulcus and middle temporal gyrus (Te3, STS1 and STS2), all surface approaches improve micro-anatomical correspondence compared to the volume alignment while differences between standard CBA and CBA+ are modest.

## Aligning in vivo group measures to the probabilistic postmortem areas

An atlas of probabilistic cytoarchitectonically defined areas has been previously used to analyze in vivo functional and anatomical data (see e.g. *Dick et al., 2012*). Here, we demonstrate the use of CBA+ and the improved version of the cytoarchitectonic atlas to this end. In particular, we aligned in vivo data collected at 7 Tesla to the CBA+ aligned postmortem cytoarchitectonic atlas. We considered only the areas in the superior temporal cortex (Te1.0, Te1.1, Te1.2, Te2.1, Te2.2 and Te3) as they were consistently included in the imaged field of view in the in vivo dataset. First, we used CBA + to produce an average morphology for the in vivo data. This alignment allowed us to derive group level maps based on the available anatomical and functional data. In particular, anatomical MRI data (0.7 mm isotropic) were used to derive intra cortical contrast related to myelin from the division of $T_1$w and $T_2^*$w data. In addition, functional MRI data (1.1 mm isotropic) collected by presenting natural sounds and analyzed with an fMRI encoding approach (*Moerel et al., 2012*), were used to derive tonotopic maps (see *Figure 10* - Supplement Figures to *Figure 10* report all the individual maps). Second, using CBA+, we aligned the average morphology of the in vivo data to the cytoarchitectonic atlas. This allowed us to project cytoarchitectonic parcels on the in vivo maps and evaluate their relationship.

Intra cortical contrast related to myelin highlights the (medial) Heschl's Gyrus as the most myelinated region in the temporal cortex (see *Figure 10*). Across cytoarchitectonic areas, Te1.0 shows the highest myelination contrast. Myelin related contrast is also high in the most medial portion of Heschl's Gyrus (Te1.1) and gradually decreases when moving away from Heschl's Gyrus.

The average tonotopic pattern highlights the Heschl's Gyrus as, for the most part, preferring low frequencies, while surrounding areas (in posterior antero-medial and antero-lateral directions) prefer high frequencies (see *Figure 10*). The high-frequency areas form an inverted 'V' pattern surrounding the Heschl's Gyrus (*Da Costa et al., 2011*; *Moerel et al., 2014*). Cytoarchitectonic primary cortical areas (Te1) cover the Heschl's Gyrus, with the core (Te1.0) in its middle section which (at the group level) appears characterized by mainly low-frequency preference (see *Figure 10*). Located medial to Te1.0, area Te1.1 may reflect an intermediate processing stage between primary and belt areas (*Moerel et al., 2014*) and covers one tonotopic gradient going from high to low in an antero-medial to postero-lateral direction. Te2.2 covers a posterior portion of the tonotopic gradient running in the posterior to anterior direction. Te2.1, covering an intermediate location between Te2.2 and Te1.0/Te1.2, overlaps with a low-frequency preferring region in the lateral portion of the Heschl's sulcus. Finally, Te3 covers a low-frequency portion of the tonotopic maps along the superior temporal gyrus (*Moerel et al., 2014*). For comparison, in a supplement to *Figure 10* we report the same maps aligned with an an atlas obtained from in vivo MRI data (using both anatomical and functional information) in a large cohort (*Glasser et al., 2016*). A direct comparison between the postmortem and in vivo atlases projected on the average anatomical curvature of our in vivo data is reported in *Figure 11*.

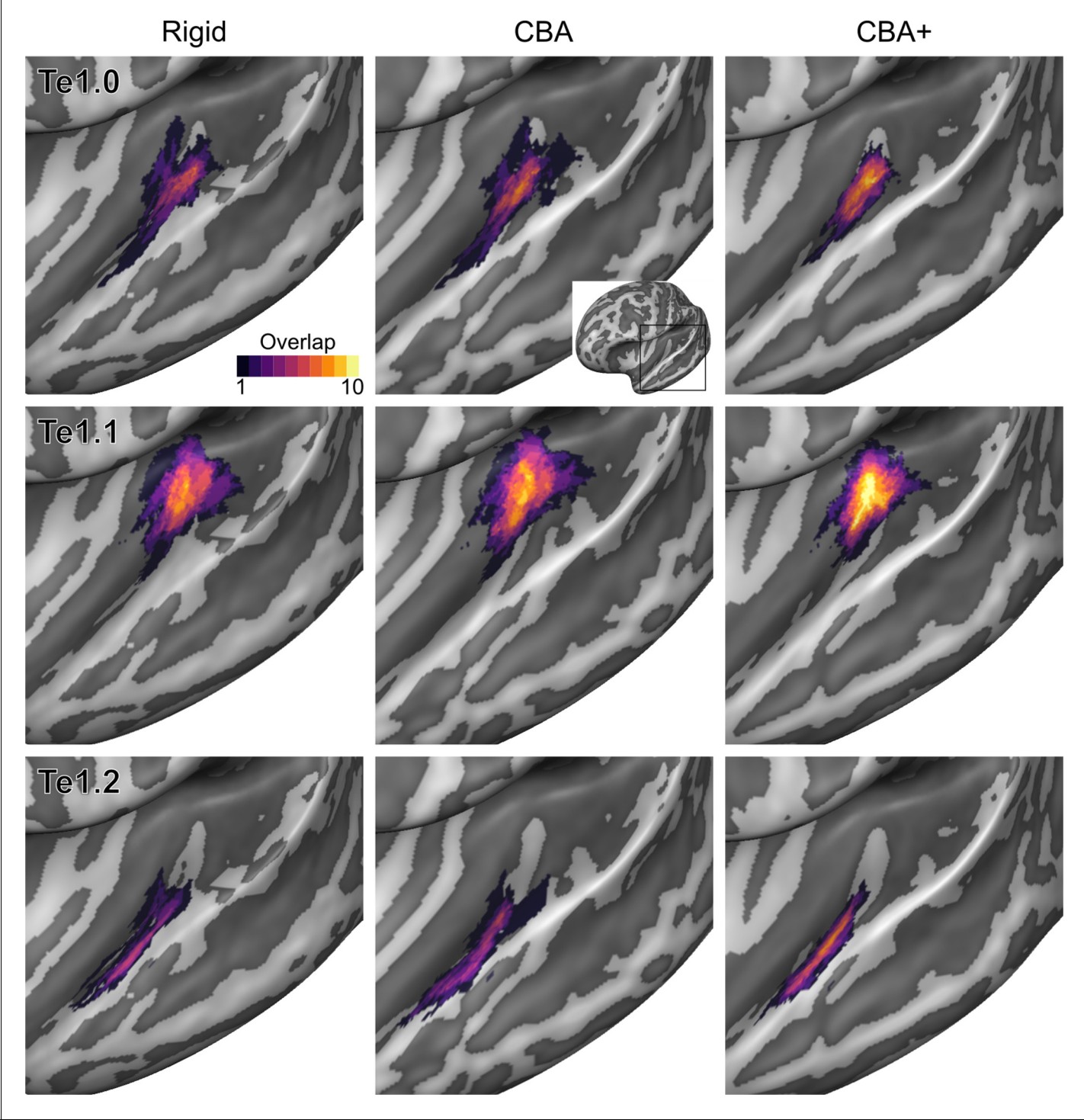

**Figure 2.** Probabilistic maps (after alignment) indicating the number of subjects for which a given vertex is labeled as belonging to the cytoarchitectonic areas Te1.0, Te1.1 and Te1.2 are presented on inflated group average cortical surfaces of the left hemisphere. Columns show spherical rigid body alignment, curvature-based alignment (CBA) and CBA with anatomical priors (CBA+) from left to right. Improvements in the micro-anatomical correspondence diminishes low values in the maps (purple) and increases the presence of high probability values (yellow).

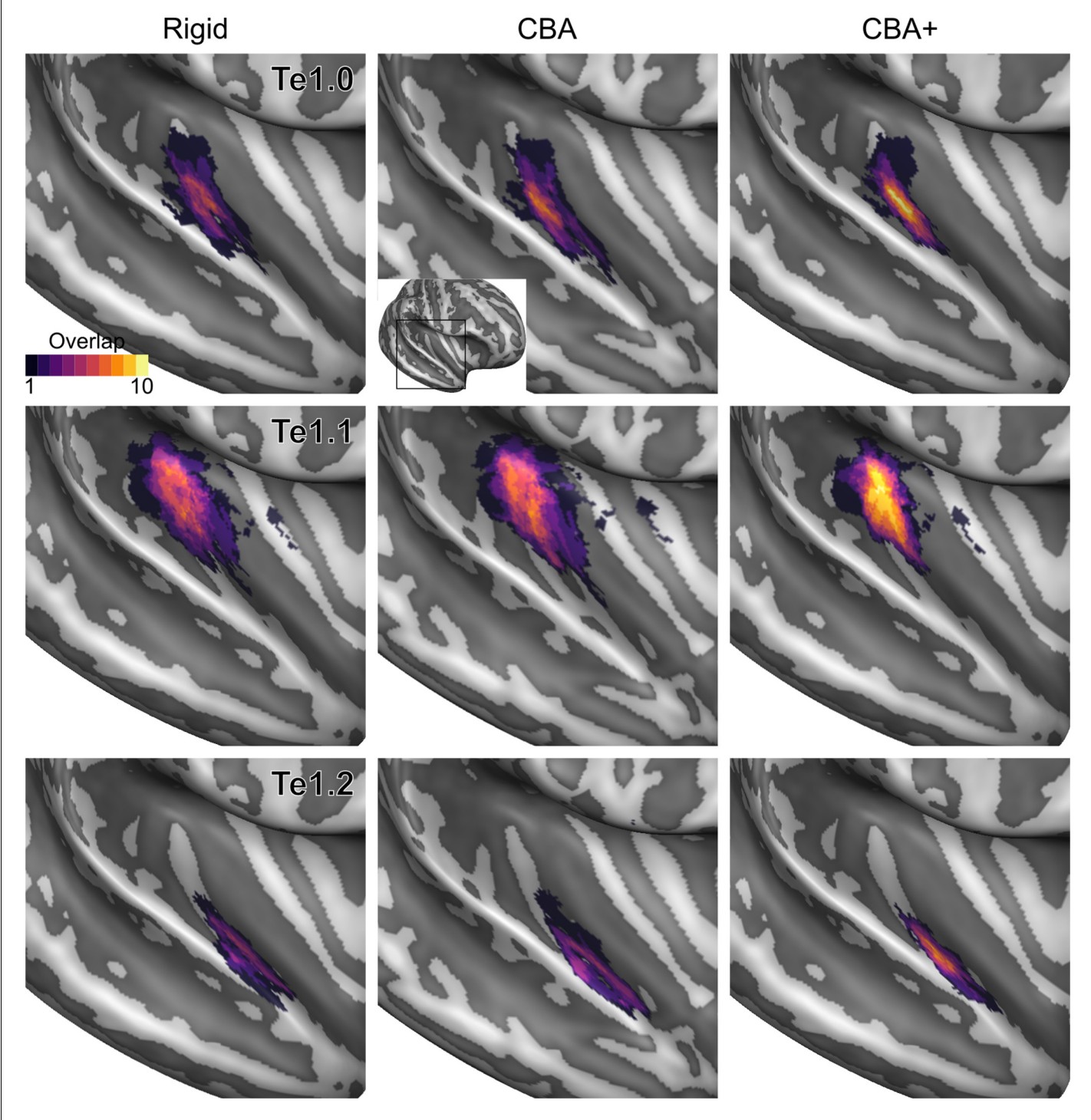

**Figure 3.** Probabilistic maps (after alignment) indicating the number of subjects for which a given vertex is labeled as belonging to the cytoarchitectonic areas Te1.0, Te1.1 and Te1.2 are presented on inflated group average cortical surfaces of the right hemisphere. Columns show spherical rigid body alignment, curvature-based alignment (CBA) and CBA with anatomical priors (CBA+) from left to right. Improvements in the micro-anatomical correspondence diminishes low values in the maps (purple) and increases the presence of high probability values (yellow).

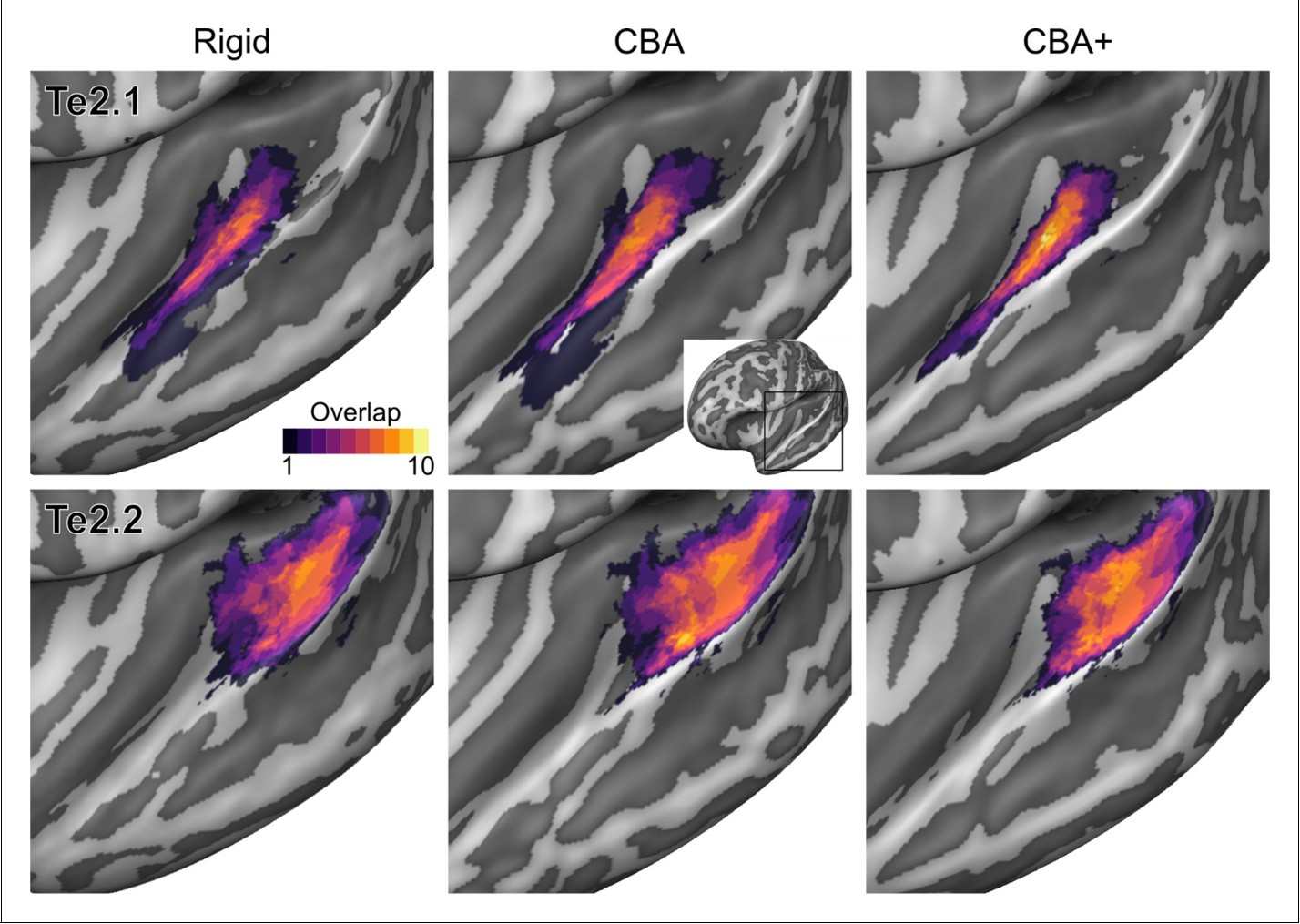

**Figure 4.** Probabilistic maps (after alignment) indicating the number of subjects for which a given vertex is labeled as belonging to the cytoarchitectonic areas Te2.1 and Te2.2 are presented on inflated group average cortical surfaces of the left hemisphere. Columns show spherical rigid body alignment, curvature-based alignment (CBA) and CBA with anatomical priors (CBA+) from left to right. Improvements the micro-anatomical correspondence diminishes low values in the maps (purple) and increases the presence of high probability values (yellow).

## Discussion

The superior temporal plane shows considerable macro-anatomical variability across individuals (*Pfeifer, 1921*; *Pfeifer, 1936*; *Von Economo and Horn, 1930*; *Rademacher et al., 1993*; *Zoellner et al., 2019*). If unaccounted for, these large inter-individual differences are detrimental to functional (and anatomical) in vivo investigations of the temporal lobe as they limit the efficacy of alignment procedures that are used in group studies. In addition, when signal-to-noise ratio (SNR) is limited, investigators rely on averaging functional (and anatomical) information, both within a subject (i.e. within a cortical area across voxels) and across subjects. This is the case for laminar (f)MRI studies conducted at high fields. While functional and anatomical localizers exist for some cortical regions in the temporal lobe (e.g., PAC -*Moerel et al., 2012*; voice regions - *Belin et al., 2000*), such localizers are not available for the majority of the auditory cortex. As a consequence, investigators often rely on available parcellations (see, e.g., the procedure followed by *Dick et al., 2012*) which result from postmortem (or in vivo) investigations in a population and as a consequence rely on the quality of the inter-subject alignment. Here, we evaluated the effect of macro-anatomical variability on localizing cytoarchitectonic areas across different brains. We have used 10 individual brains available from the JuBrain cytoarchitectonic Atlas (The JuBrain atlas is available through the Atlas of the Human Brain Project https://jubrain.fz-juelich.de/ (*Amunts and Zilles, 2015*) together with a

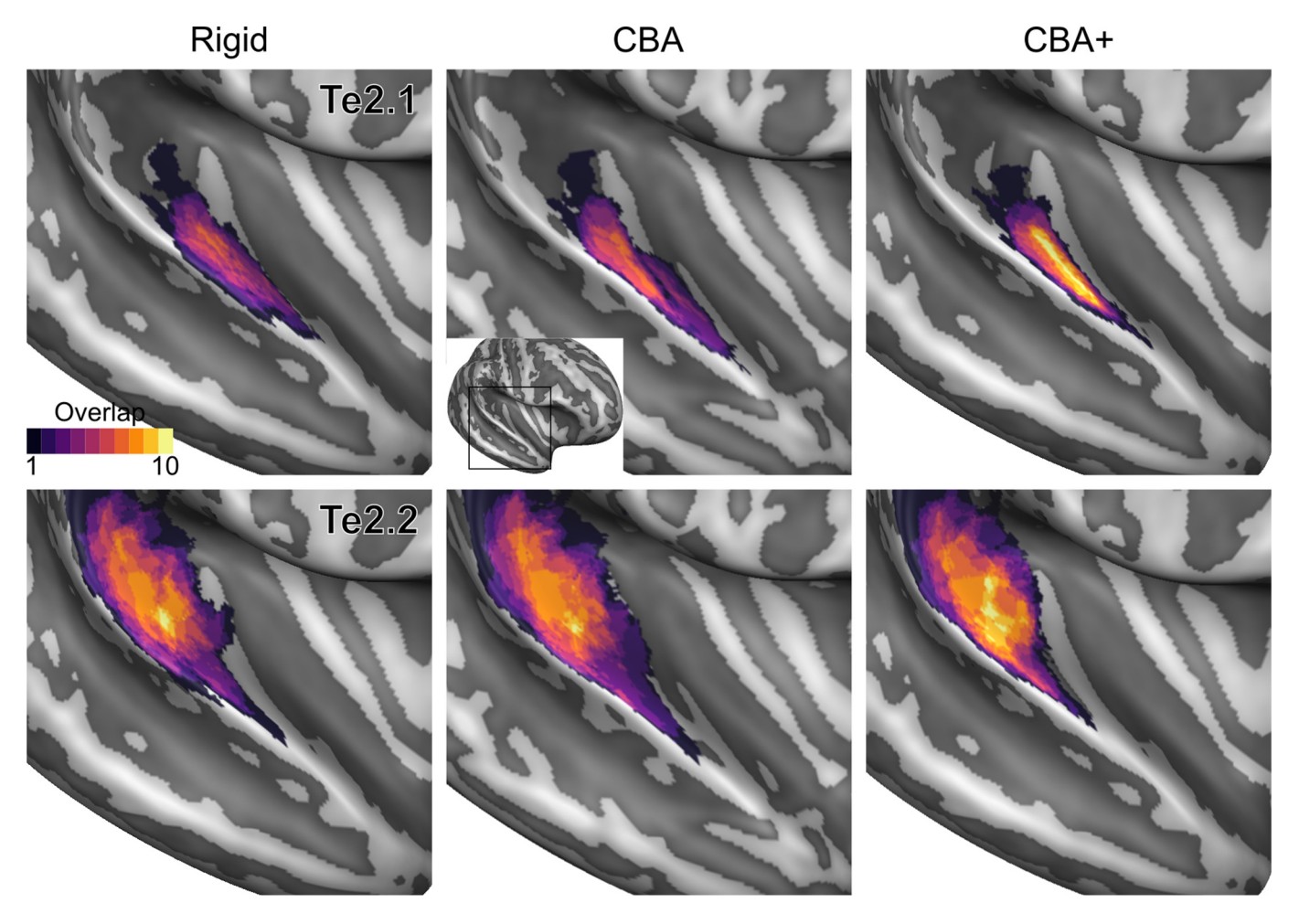

**Figure 5.** Probabilistic maps (after alignment) indicating the number of subjects for which a given vertex is labeled as belonging to the cytoarchitectonic areas Te2.1 and Te2.2 are presented on inflated group average cortical surfaces of the right hemisphere. Columns show spherical rigid body alignment, curvature-based alignment (CBA) and CBA with anatomical priors (CBA+) from left to right. Improvements in the micro-anatomical correspondence diminishes low values in the maps (purple) and increases the presence of high probability values (yellow).

surface registration method that minimizes macro-anatomical variability around the transverse temporal gyrus (similar to *Rosenke et al., 2018*) to show that minimizing macro-anatomical variability in the superior temporal plane results in improved micro-anatomical correspondence across brains. Our results have two potential benefits for the auditory neuroscience community. First, by providing a more accurate (i.e. with improved inter subject alignment) probabilistic atlas of the cytoarchitectonic areas in the temporal lobe, we provide a valuable resource for those studies that rely on a parcellation scheme. Second, we show that minimizing a particular set of macro-anatomical features results in a more accurate micro-anatomical alignment, and thus that using local landmarks for aligning temporal regions results in a better alignment of cytoarchitectonic areas. This justifies the use of these approaches (the one we propose here or others that have been proposed before *Kang et al., 2004*; *Desai et al., 2005*) when aligning individual temporal cortices with each other (or to a template).

Applying a surface registration for inter-subject alignment required accurate segmentation of the postmortem MRI dataset. While this issue has been tackled before for the investigation of cytoarchitectonic areas in the visual cortex *Rosenke et al., 2018*, an accurate segmentation of the temporal areas was not available. To obtain such segmentation and reduce the amount of manual corrections, we have used a tailored procedure based on image filtering and histogram-based segmentation

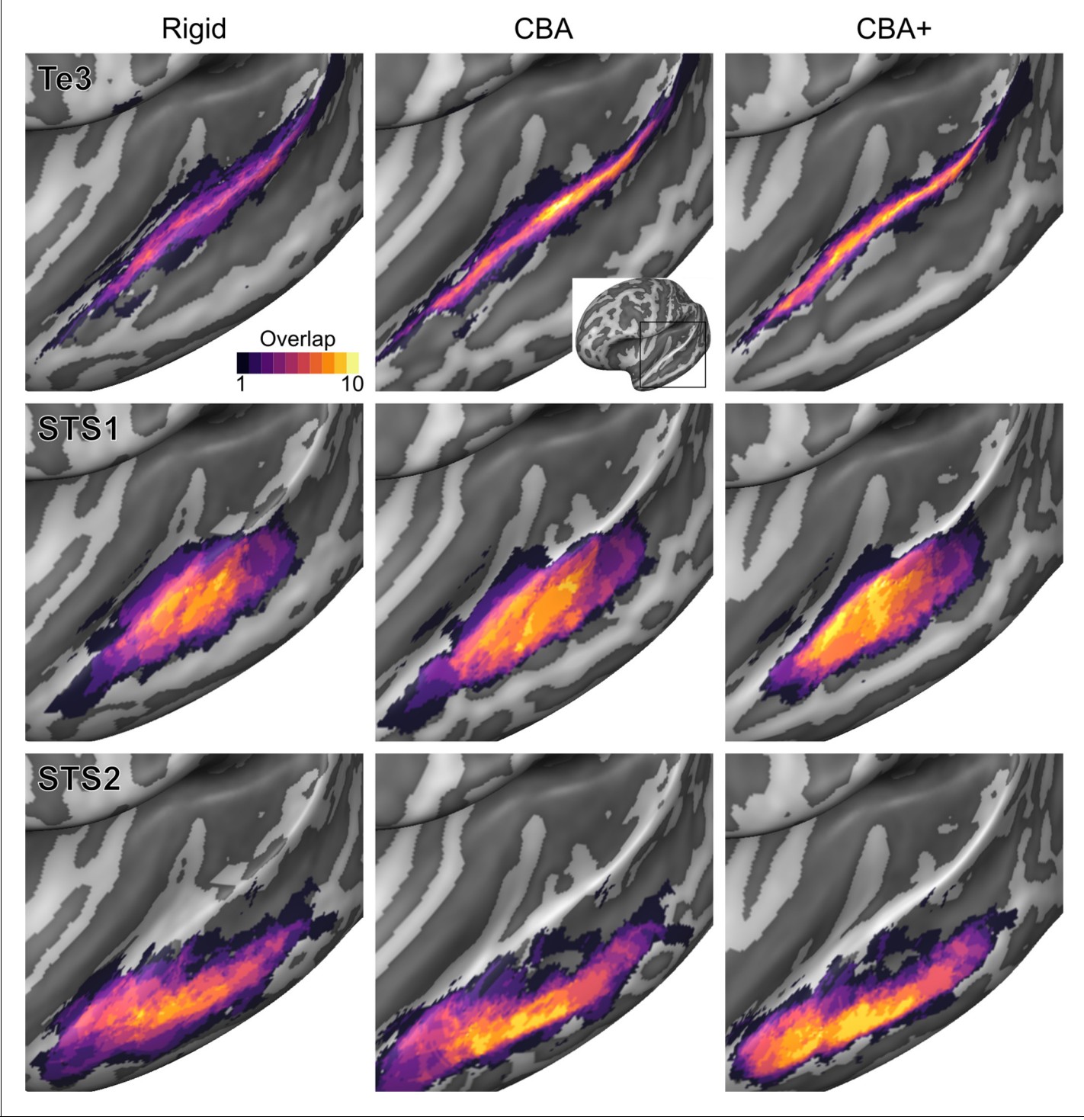

**Figure 6.** Probabilistic maps (after alignment) indicating the number of subjects for which a given vertex is labeled as belonging to the cytoarchitectonic areas Te3, STS1 and STS2 are presented on inflated group average cortical surfaces of the left hemisphere. Columns show spherical rigid body alignment, curvature-based alignment (CBA) and CBA with anatomical priors (CBA+) from left to right. Improvements in the micro-anatomical correspondence diminishes low values in the maps (purple) and increases the presence of high probability values (yellow).

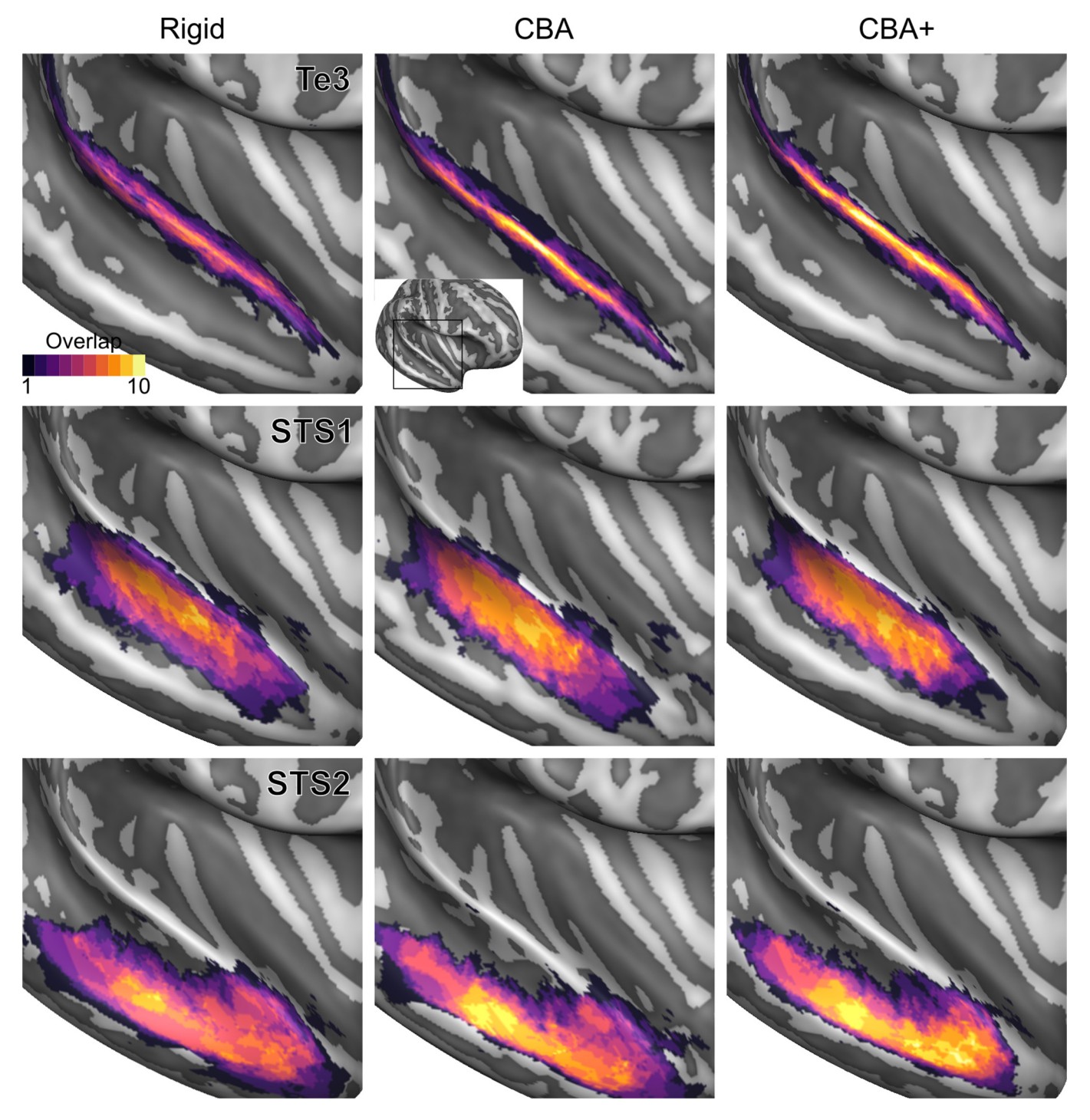

**Figure 7.** Probabilistic maps (after alignment) indicating the number of subjects for which a given vertex is labeled as belonging to the cytoarchitectonic areas Te3, STS1 and STS2 are presented on inflated group average cortical surfaces of the right hemisphere. Columns show spherical rigid body alignment, curvature-based alignment (CBA) and CBA with anatomical priors (CBA+) from left to right. Improvements in the micro-anatomical correspondence diminishes low values in the maps (purple) and increases the presence of high probability values (yellow).

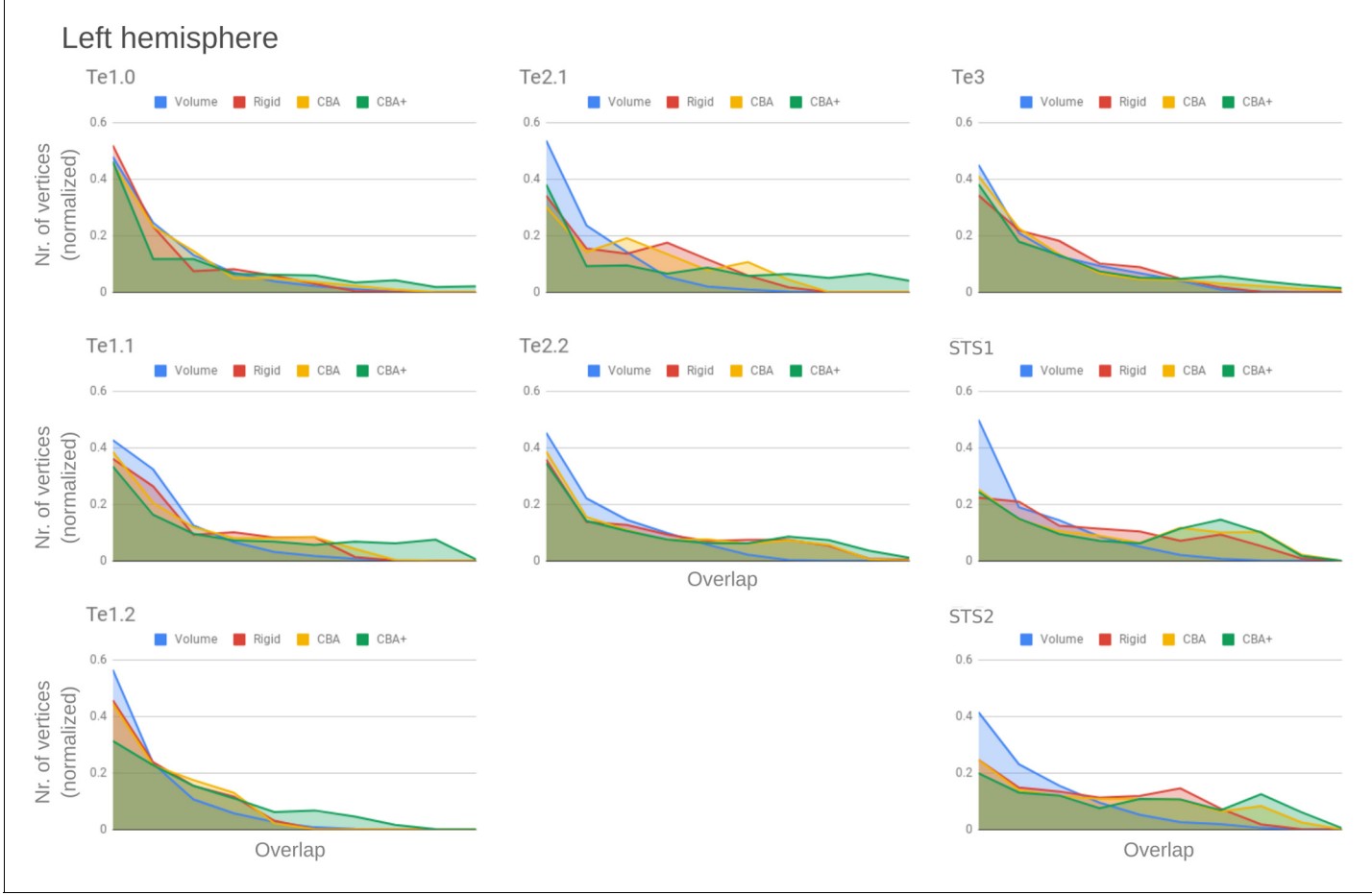

**Figure 8.** Histograms of the overlap across cytoarchitectonic areas in the left hemisphere. The histograms are normalized by the number of vertices per area. The x-axis represents the probability value (an overlap from 1 out of ten [left] to 10 out of 10 participants [right]). The ideal co-registration method should show a less left skewed distribution. CBA+ shows the lowest skew towards the left in comparison to other methods.

(*Gulban et al., 2018b*). The resulting segmentations allowed us to define the macro-anatomical variability in the sample (see Figure 13). The availabe 10 brains showed typical variations in the morphology of the Heschl's Gyrus (with a single Heschl's Gyrus being the most prevalent one), as well as cases in which the Heschl's Gyrus was continuous to the anterior portion of the superior temporal gyrus (*Heschl, 1878*).

The segmented hemispheres were used for cortex-based alignment. The standard approach minimizes macro-anatomical variation across subjects (i.e. maximizes the overlap of the curvature maps) across the whole brain (in a coarse to fine iterative approach). As such, standard CBA is driven by the major anatomical landmarks including the superior temporal gyrus and sulcus. This, however, can result in compromised alignment of smaller (but consistent) anatomical features such as the Heschl's Gyrus. This can be seen in *Figure 1* (middle column) where the compromised alignment of the Heschl's Gyrus across hemispheres is indicated by the reduced sharpness of the averaged binarized curvature maps. For this reason, here we have considered the application of an approach tailored to the superior temporal plane. The necessity to use local landmarks to guide the alignment of temporal regions has been considered in previous research (*Kang et al., 2004*; *Desai et al., 2005*). By defining macro-anatomical points in the temporal cortex (*Kang et al., 2004*) or lines covering the main gyri and sulci (*Desai et al., 2005*) both linear and non-linear alignment procedures have been compared to global spherical alignment and volumetric approaches. Here, we provided additional landmarks (the Heschl's Gyrus, the superior temporal gyrus/sulcus and middle temporal gyrus) to the CBA procedure and, differently from previous approaches (that considered only the local landmarks when aligning the temporal cortex), we optimized alignment of both local and global macro-

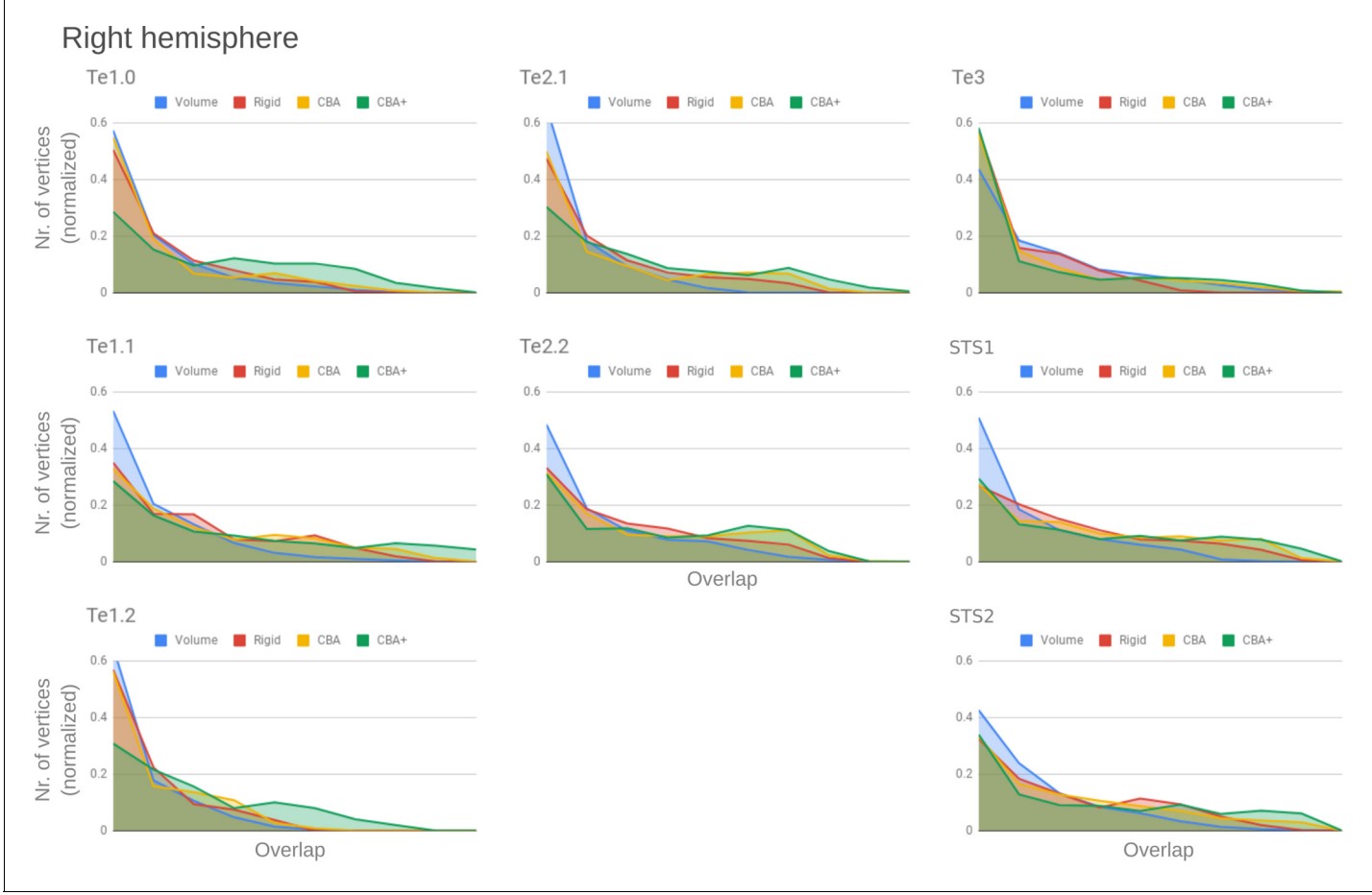

**Figure 9.** Histograms of the overlap across cytoarchitectonic areas in the right hemisphere. The histograms are normalized by the number of vertices per area. The x-axis represents the probability value (an overlap from 1 out of 10 [left] to 10 out of 10 participants [right]). The ideal co-registration method should show less left skewed distribution. CBA+ shows the lowest skew toward the left in comparison to other methods.

anatomical features. By doing so, we improved the alignment across subjects in the superior temporal cortex (see e.g. the difference in the average curvature maps between standard CBA and CBA+ in *Figure 1*). Both the CBA and CBA+ approach greatly improved the macro-anatomical correspondence when compared to a rigid body procedure (which by sampling the volumetric data on surfaces already offers an improvement compared to the original volumetric alignment - see *Figure 8* and *Figure 9*). This result is in line with the previously reported increased macro-anatomical correspondence obtained when using alignment procedures that use local landmarks (*Kang et al., 2004*; *Desai et al., 2005*). The advantage of the tailored approach (CBA+, rightmost column in *Figure 1*) is stronger in the right hemisphere, with some residual misalignment for the left Heschl's Gyrus. This difference in performance could be explained by the larger prevalence (within our sample) in the left hemisphere of cases with the Heschl's Gyrus merging with the anterior portion of the superior temporal gyrus (i.e. split superior temporal gyrus cases; two in the left and one in the right hemisphere). In the future, a larger sample could allow evaluating this issue, as well as the impact that the inclusion of this macro-anatomical variation has on the alignment of regions close to the superior temporal gyrus, by evaluating the alignment separately (with and without) such cases.

Tailoring the alignment of the temporal lobe to local landmarks is motivated by the expectation that minimizing macro-anatomy will consequently result in improved micro-anatomical alignment (i.e. the alignment of cortical areas in the temporal lobe) and that this in turn improves functional alignment (assuming that function and micro-anatomy co-localize). While previous studies have shown that approaches that use local landmarks result in improved group functional activation (*Kang et al., 2004*; *Desai et al., 2005*), here we addressed the much required validation of how well

**Table 1.** DICE coefficients computed for each area using a leave-one-subject out approach for the areas in left hemispheres. Using rigid body alignment, CBA, or CBA+, the surface area of each area from the left-out subject is compared to an atlas obtained from the remaining nine subjects. The atlas in this case is constructed by considering the vertices that overlap in at least four out of the nine subjects.

**RIGID**

| Area | Sub-01 | Sub-03 | Sub-04 | Sub-05 | Sub-06 | Sub-07 | Sub-08 | Sub-09 | Sub-10 | Sub-13 | Mean | Std. Dev. |
|---|---|---|---|---|---|---|---|---|---|---|---|---|
| Te 1.0 | 0.23 | 0.47 | 0.42 | 0.36 | 0.40 | 0.00 | 0.43 | 0.45 | 0.02 | 0.12 | 0.29 | 0.18 |
| Te 1.1 | 0.50 | 0.71 | 0.54 | 0.62 | 0.62 | 0.09 | 0.67 | 0.49 | 0.33 | 0.15 | 0.47 | 0.21 |
| Te 1.2 | 0.03 | 0.40 | 0.02 | 0.25 | 0.43 | 0.00 | 0.00 | 0.18 | 0.23 | 0.00 | 0.15 | 0.17 |
| Te 2.1 | 0.56 | 0.42 | 0.43 | 0.49 | 0.70 | 0.00 | 0.34 | 0.40 | 0.22 | 0.22 | 0.38 | 0.20 |
| Te 2.2 | 0.36 | 0.49 | 0.57 | 0.64 | 0.57 | 0.49 | 0.62 | 0.55 | 0.10 | 0.09 | 0.45 | 0.20 |
| Te 3 | 0.26 | 0.11 | 0.08 | 0.23 | 0.35 | 0.01 | 0.20 | 0.35 | 0.15 | 0.27 | 0.20 | 0.11 |
| Te 4 | 0.57 | 0.41 | 0.61 | 0.54 | 0.64 | 0.27 | 0.34 | 0.40 | 0.70 | 0.64 | 0.51 | 0.15 |
| Te 5 | 0.54 | 0.41 | 0.65 | 0.45 | 0.69 | 0.43 | 0.29 | 0.57 | 0.61 | 0.54 | 0.52 | 0.12 |

**CBA**

| Area | Sub-01 | Sub-03 | Sub-04 | Sub-05 | Sub-06 | Sub-07 | Sub-08 | Sub-09 | Sub-10 | Sub-13 | Mean | Std. Dev. |
|---|---|---|---|---|---|---|---|---|---|---|---|---|
| Te 1.0 | 0.22 | 0.62 | 0.65 | 0.61 | 0.57 | 0.03 | 0.39 | 0.30 | 0.25 | 0.59 | 0.42 | 0.21 |
| Te 1.1 | 0.57 | 0.75 | 0.57 | 0.70 | 0.66 | 0.25 | 0.73 | 0.35 | 0.36 | 0.46 | 0.54 | 0.18 |
| Te 1.2 | 0.01 | 0.04 | 0.25 | 0.22 | 0.14 | 0.11 | 0.00 | 0.07 | 0.24 | 0.18 | 0.12 | 0.09 |
| Te 2.1 | 0.82 | 0.54 | 0.56 | 0.61 | 0.84 | 0.00 | 0.35 | 0.21 | 0.58 | 0.70 | 0.52 | 0.26 |
| Te 2.2 | 0.46 | 0.60 | 0.65 | 0.75 | 0.67 | 0.68 | 0.68 | 0.57 | 0.19 | 0.27 | 0.55 | 0.19 |
| Te 3 | 0.59 | 0.34 | 0.62 | 0.56 | 0.59 | 0.34 | 0.30 | 0.60 | 0.46 | 0.51 | 0.49 | 0.12 |
| Te 4 | 0.68 | 0.58 | 0.72 | 0.57 | 0.69 | 0.41 | 0.35 | 0.45 | 0.75 | 0.68 | 0.59 | 0.14 |
| Te 5 | 0.59 | 0.47 | 0.72 | 0.47 | 0.66 | 0.53 | 0.22 | 0.53 | 0.53 | 0.52 | 0.52 | 0.13 |

**CBA+**

| Area | Sub-01 | Sub-03 | Sub-04 | Sub-05 | Sub-06 | Sub-07 | Sub-08 | Sub-09 | Sub-10 | Sub-13 | Mean | Std |
|---|---|---|---|---|---|---|---|---|---|---|---|---|
| Te 1.0 | 0.18 | 0.74 | 0.61 | 0.49 | 0.66 | 0.68 | 0.50 | 0.65 | 0.45 | 0.59 | 0.56 | 0.16 |
| Te 1.1 | 0.65 | 0.71 | 0.58 | 0.62 | 0.68 | 0.65 | 0.72 | 0.72 | 0.64 | 0.40 | 0.64 | 0.10 |
| Te 1.2 | 0.38 | 0.24 | 0.42 | 0.52 | 0.66 | 0.60 | 0.09 | 0.58 | 0.55 | 0.60 | 0.47 | 0.18 |
| Te 2.1 | 0.69 | 0.55 | 0.53 | 0.61 | 0.78 | 0.42 | 0.40 | 0.55 | 0.61 | 0.61 | 0.57 | 0.11 |
| Te 2.2 | 0.48 | 0.66 | 0.62 | 0.67 | 0.69 | 0.71 | 0.67 | 0.68 | 0.37 | 0.23 | 0.58 | 0.16 |
| Te 3 | 0.60 | 0.39 | 0.62 | 0.65 | 0.58 | 0.64 | 0.34 | 0.53 | 0.58 | 0.38 | 0.53 | 0.12 |
| Te 4 | 0.76 | 0.82 | 0.75 | 0.52 | 0.66 | 0.56 | 0.35 | 0.49 | 0.79 | 0.68 | 0.64 | 0.15 |
| Te 5 | 0.71 | 0.65 | 0.79 | 0.64 | 0.68 | 0.59 | 0.33 | 0.54 | 0.68 | 0.68 | 0.63 | 0.13 |

such an approach reduces the underlying micro-anatomical variability. Improving macro-anatomical correspondence resulted in improved overlap of the cytoarchitectonic areas across subjects. As a result of the CBA+ alignment, the micro-anatomically defined areas were smaller and the probability for a vertex to be labeled as belonging to the same area across the postmortem samples was higher (see *Figures 2–7* and the histograms in *Figure 8* and *Figure 9*). The tailored approach (CBA+) resulted in increased overlap (also compared to standard CBA) in all areas but especially in those on Heschl's Gyrus or immediately adjacent to it (Te1.0, Te1.1, Te1.2 and Te2.1 - see *Table 1* and *Table 2*). This result is a direct consequence of defining the (most anterior) Heschl's Gyrus as an additional landmark for alignment. The most anterior Heschl's Gyrus was recognized as the putative location of primary auditory cortex in the case of a complete duplication on the basis of myelo-architecture (*Hackett et al., 2001*). When this anatomical landmark is not used, the duplication of the Heschl's Gyrus results in poorer matching across subjects (i.e. the most posterior duplication of some subjects is aligned to the single Heschl's Gyrus of other subjects). The postmortem dataset

**Table 2.** DICE coefficients computed for each area using a leave-one-subject out approach for the areas in right hemispheres. Using rigid body alignment, CBA, or CBA+, the surface area of each area from the left-out subject is compared to an atlas obtained from the remaining nine subjects. The atlas in this case is constructed by considering the vertices that overlap in at least four out of the nine subjects.

**RIGID**

| Area | Sub-01 | Sub-03 | Sub-04 | Sub-05 | Sub-06 | Sub-07 | Sub-08 | Sub-09 | Sub-10 | Sub-13 | Mean | Std. Dev. |
|------|--------|--------|--------|--------|--------|--------|--------|--------|--------|--------|------|-----------|
| Te 1.0 | 0.08 | 0.58 | 0.09 | 0.32 | 0.25 | 0.56 | 0.44 | 0.31 | 0.25 | 0.09 | 0.30 | 0.18 |
| Te 1.1 | 0.56 | 0.57 | 0.34 | 0.36 | 0.06 | 0.51 | 0.51 | 0.55 | 0.34 | 0.18 | 0.40 | 0.17 |
| Te 1.2 | 0.00 | 0.08 | 0.00 | 0.25 | 0.24 | 0.18 | 0.00 | 0.13 | 0.14 | 0.00 | 0.10 | 0.10 |
| Te 2.1 | 0.20 | 0.38 | 0.37 | 0.36 | 0.52 | 0.49 | 0.52 | 0.34 | 0.18 | 0.12 | 0.35 | 0.14 |
| Te 2.2 | 0.64 | 0.46 | 0.71 | 0.65 | 0.64 | 0.65 | 0.65 | 0.51 | 0.30 | 0.30 | 0.55 | 0.15 |
| Te 3 | 0.29 | 0.16 | 0.30 | 0.35 | 0.48 | 0.53 | 0.42 | 0.29 | 0.42 | 0.48 | 0.37 | 0.12 |
| Te 4 | 0.76 | 0.54 | 0.58 | 0.59 | 0.16 | 0.64 | 0.57 | 0.59 | 0.68 | 0.29 | 0.54 | 0.18 |
| Te 5 | 0.62 | 0.35 | 0.72 | 0.58 | 0.15 | 0.68 | 0.58 | 0.54 | 0.74 | 0.42 | 0.54 | 0.18 |

**CBA**

| Area | Sub-01 | Sub-03 | Sub-04 | Sub-05 | Sub-06 | Sub-07 | Sub-08 | Sub-09 | Sub-10 | Sub-13 | Mean | Std. Dev. |
|------|--------|--------|--------|--------|--------|--------|--------|--------|--------|--------|------|-----------|
| Te 1.0 | 0.29 | 0.72 | 0.47 | 0.39 | 0.42 | 0.60 | 0.61 | 0.20 | 0.12 | 0.00 | 0.38 | 0.23 |
| Te 1.1 | 0.66 | 0.58 | 0.47 | 0.47 | 0.26 | 0.63 | 0.56 | 0.48 | 0.30 | 0.14 | 0.46 | 0.17 |
| Te 1.2 | 0.00 | 0.08 | 0.00 | 0.16 | 0.15 | 0.12 | 0.00 | 0.07 | 0.09 | 0.03 | 0.07 | 0.06 |
| Te 2.1 | 0.47 | 0.54 | 0.72 | 0.32 | 0.58 | 0.59 | 0.67 | 0.22 | 0.01 | 0.01 | 0.41 | 0.26 |
| Te 2.2 | 0.70 | 0.45 | 0.77 | 0.70 | 0.65 | 0.67 | 0.66 | 0.53 | 0.24 | 0.28 | 0.56 | 0.18 |
| Te 3 | 0.56 | 0.39 | 0.49 | 0.42 | 0.47 | 0.69 | 0.44 | 0.25 | 0.51 | 0.51 | 0.47 | 0.11 |
| Te 4 | 0.84 | 0.76 | 0.68 | 0.63 | 0.09 | 0.67 | 0.63 | 0.76 | 0.79 | 0.33 | 0.62 | 0.23 |
| Te 5 | 0.71 | 0.54 | 0.74 | 0.53 | 0.10 | 0.64 | 0.65 | 0.77 | 0.76 | 0.43 | 0.59 | 0.20 |

**CBA+**

| Area | Sub-01 | Sub-03 | Sub-04 | Sub-05 | Sub-06 | Sub-07 | Sub-08 | Sub-09 | Sub-10 | Sub-13 | Mean | Std |
|------|--------|--------|--------|--------|--------|--------|--------|--------|--------|--------|------|-----|
| Te 1.0 | 0.54 | 0.68 | 0.60 | 0.32 | 0.62 | 0.66 | 0.70 | 0.64 | 0.55 | 0.51 | 0.58 | 0.11 |
| Te 1.1 | 0.77 | 0.71 | 0.60 | 0.40 | 0.66 | 0.68 | 0.57 | 0.71 | 0.66 | 0.64 | 0.64 | 0.10 |
| Te 1.2 | 0.45 | 0.58 | 0.47 | 0.21 | 0.60 | 0.57 | 0.30 | 0.33 | 0.48 | 0.35 | 0.43 | 0.13 |
| Te 2.1 | 0.66 | 0.66 | 0.82 | 0.48 | 0.75 | 0.68 | 0.76 | 0.67 | 0.63 | 0.50 | 0.66 | 0.11 |
| Te 2.2 | 0.74 | 0.47 | 0.73 | 0.72 | 0.65 | 0.67 | 0.66 | 0.70 | 0.47 | 0.49 | 0.63 | 0.11 |
| Te 3 | 0.57 | 0.55 | 0.50 | 0.44 | 0.47 | 0.70 | 0.48 | 0.62 | 0.61 | 0.65 | 0.56 | 0.09 |
| Te 4 | 0.82 | 0.83 | 0.64 | 0.70 | 0.25 | 0.67 | 0.66 | 0.78 | 0.77 | 0.37 | 0.65 | 0.19 |
| Te 5 | 0.75 | 0.82 | 0.77 | 0.61 | 0.39 | 0.63 | 0.76 | 0.82 | 0.60 | 0.53 | 0.67 | 0.14 |

includes six Heschl's Gyrus duplication cases (four in the right and two in the left hemisphere). Follow-up studies are needed to evaluate the effect of an incomplete duplication of Heschl's Gyrus. As previous myelo-architecture studies reported a shift of primary areas toward the intermediate Heschl's sulcus in the case of an incomplete duplication (*Hackett et al., 2001*), a partial alignment of the primary areas (Te1.0 and Te1.1) may be expected. Examining the effect of an incomplete duplication on micro-anatomical alignment may provide additional insights for a further refinement of the alignment procedure we propose here. In addition to the anterior Heschl's Gyrus, CBA+ includes the superior temporal gyrus/sulcus and middle temporal gyrus as anatomical landmarks. While to a lesser degree than the areas on Heschl's Gyrus, areas along these landmarks were also better realigned by CBA+. This indicates that favoring these gyri/sulci with respect to other major landmarks on the cortex is beneficial for the alignment of temporal areas. The improved cytoarchitectonic overlap obtained with CBA+ suggests that this approach may be relevant for the functional and anatomical investigation of (auditory) temporal areas in vivo, as well as the investigation (postmortem and in

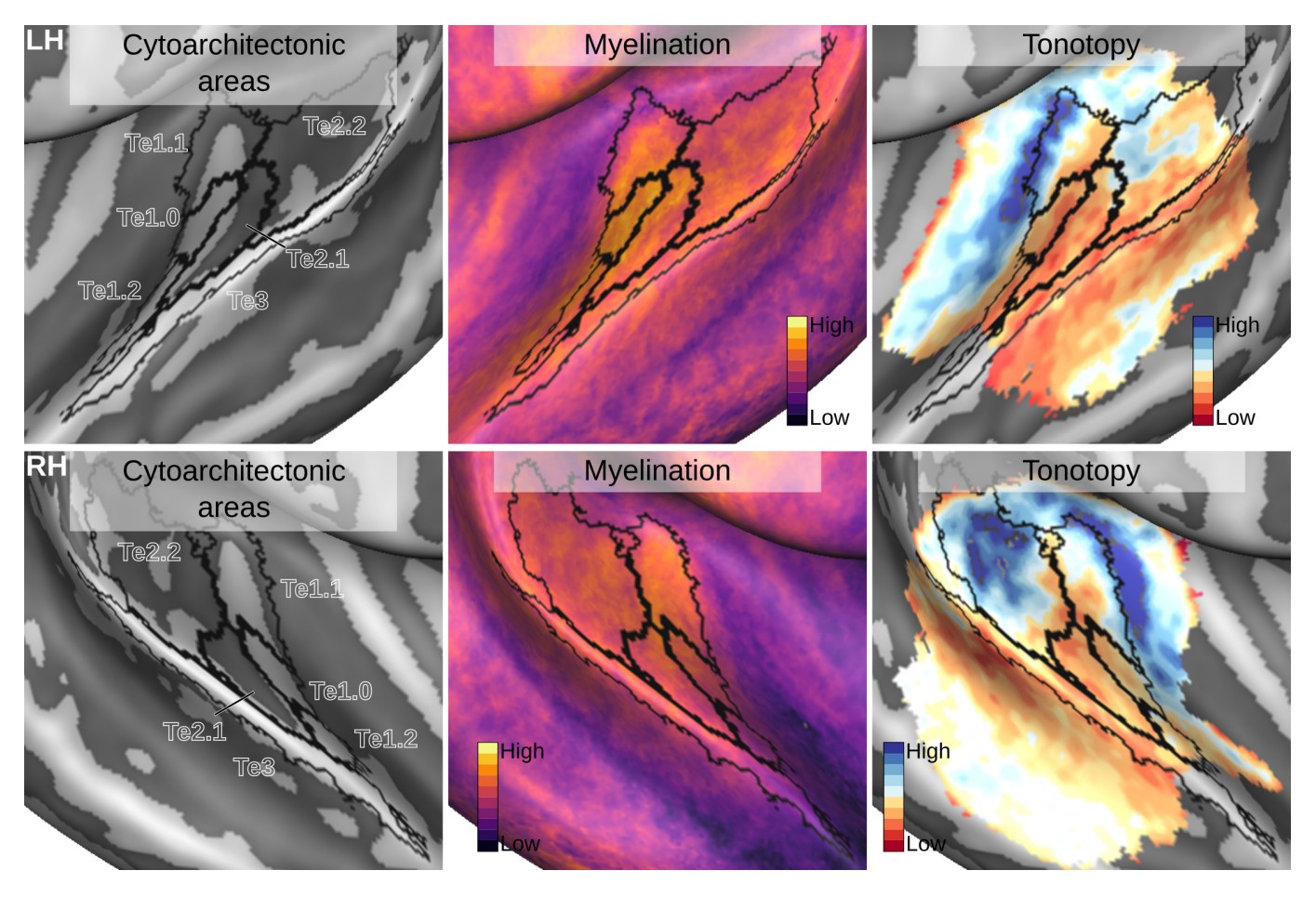

**Figure 10.** Relation between group average (CBA+) in vivo MRI measures and the cytoarchitectonic atlas. The cytoarchitectonic areas are delineated with black lines. The myelination index is computed from the division of $T_1$w and $T_2^*$w data. Tonotopy reflects the voxel-wise frequency preference estimated with fMRI encoding from the response to natural sound stimuli. All measures are sampled on the middle gray matter surfaces. The online version of this article includes the following figure supplement(s) for figure 10:

**Figure supplement 1.** The same maps projected on an in vivo multi-modal MRI group atlas (*Glasser et al., 2016*).
**Figure supplement 2.** Individual brain in vivo MRI measures of Subject 01.
**Figure supplement 3.** Individual brain in vivo MRI measures of Subject 02.
**Figure supplement 4.** Individual brain in vivo MRI measures of Subject 03.
**Figure supplement 5.** Individual brain in vivo MRI measures of Subject 05.
**Figure supplement 6.** Individual brain in vivo MRI measures of Subject 06.
**Figure supplement 7.** Individual brain in vivo MRI measures of Subject 07.
**Figure supplement 8.** Individual brain in vivo MRI measures of Subject 08.
**Figure supplement 9.** Individual brain in vivo MRI measures of Subject 09.
**Figure supplement 10.** Individual brain in vivo MRI measures of Subject 10.
**Figure supplement 11.** Individual brain in vivo MRI measures of Subject 11.

vivo ) of other cortical regions in which macro-anatomical variability is high. We make the individual hemisphere surface models and the individual cytoarchitectonic areas used in this study publicly available at https://kg.ebrains.eu/search/instances/Dataset/ff71a4d1-ea14-4ed6-898e-b92d95b3c446. In follow-up investigations, this publicly available data set could also be used to compare the performance of different landmark based approaches that use only local landmarks (*Kang et al., 2004*; *Desai et al., 2005*) as well as non-linear volumetric approaches to CBA+.

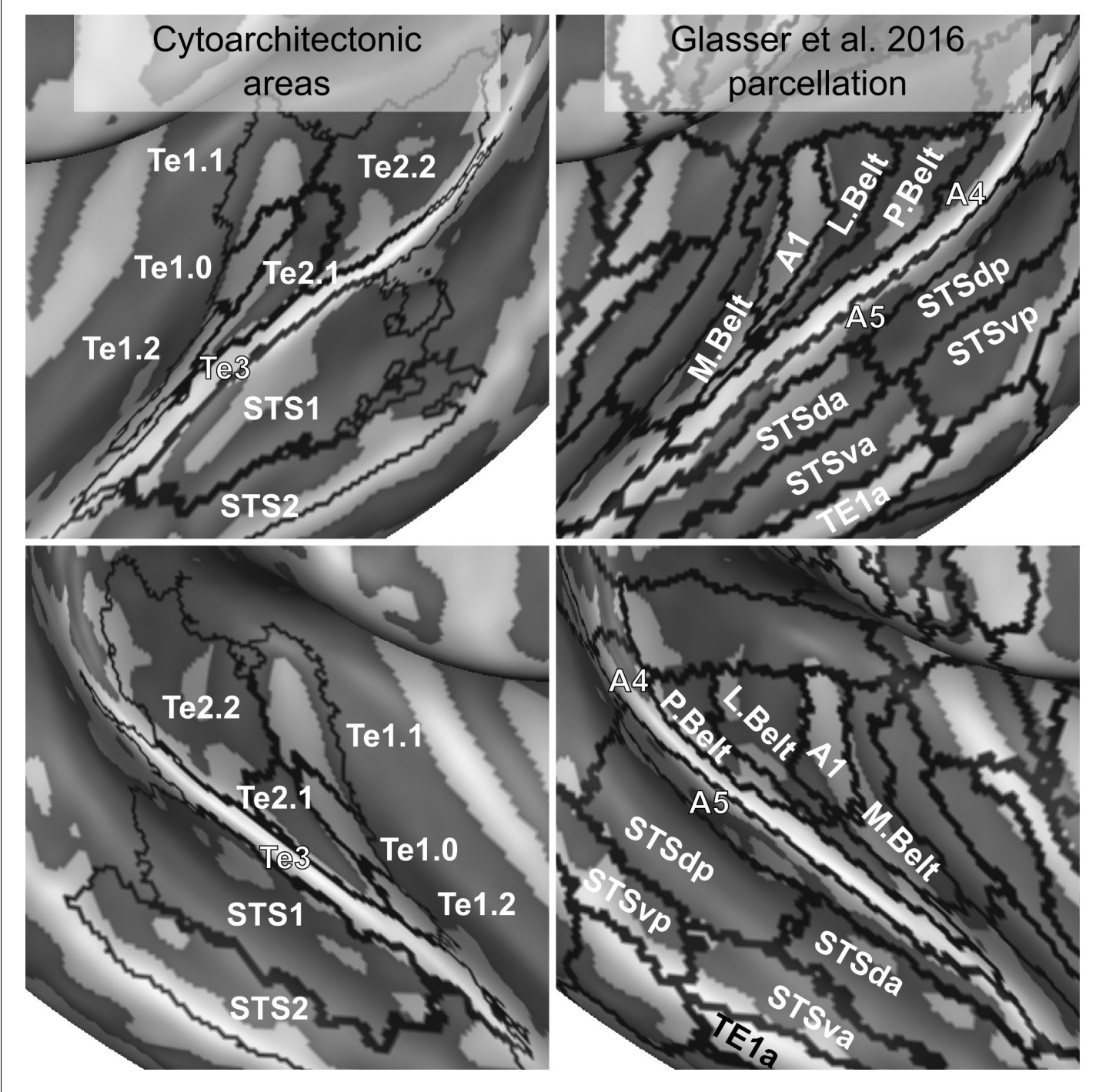

**Figure 11.** Comparison of cytoarchitectonic areas *Morosan et al., 2001*; *Morosan et al., 2005* and multi-modal MRI-based labels (*Glasser et al., 2016*). Areas on Heschl's Gyrus differ between the two atlases.

To showcase the application of CBA+ to the analysis of in vivo MRI data, we applied the same procedure to align anatomical and functional data collected at 7 Tesla across individuals. In addition, we used CBA+ to align the in vivo data to the improved cytoarchitectonic atlas.

The pattern of myelin related intra cortical contrast followed previous reports (*Glasser and Van Essen, 2011*; *Dick et al., 2012*; *De Martino et al., 2015*). The alignment to the cytoarchitectonic atlas shows a high myelin-related contrast in area Te1.0, in agreement with previous studies

(*Dick et al., 2012*). Myelin-related contrast was high also in the most medial portion of Heschl's Gyrus (Te1.1) and decreased when moving away from Heschl's Gyrus. While subtle differences between Te1.0 and Te1.1 were already noticeable, a more clear cut separation between these regions may require the evaluation of myelin-related contrast across depths similarly to previous approaches (*Dick et al., 2012*; *De Martino et al., 2015*). In addition, future investigations may evaluate the information provided by intra anatomical contrast resulting from in vivo MRI acquisitions other than the one we considered here (e.g. using the orientation of intra-cortical fibers [*McNab et al., 2013*; *Gulban et al., 2018a*]).

The group tonotopy maps we derived from the in vivo data follow previous reports (*Dick et al., 2012*; *Moerel et al., 2014*; *Besle et al., 2018*). In particular, they show one gradient within area Te1.1 progressing from high to low frequencies in antero-medial to postero-lateral direction. Based on the average maps, a full tonotopic gradient was not visible in Te1.0, which was corresponding mainly with the low frequency area on medial Heschl's Gyrus. This pattern may be the result of excessive smoothing caused by inter-subject averaging that highlights the larger frequency gradient that in tonotopic maps progresses in the anterior-posterior direction on the planum temporale and thus favors the interpretation of the pattern within larger cortical areas (*Moerel et al., 2014*). While future studies are needed to quantify the information that can be leveraged from individual in vivo MRI data for characterizing the cytoarchitectonic areas, our preliminary results indicate that individual tonotopic [and myelin] maps (supplement figures to *Figure 10*) provide more fine grained information (within smaller areas such as e.g. Te1.0) (*Moerel et al., 2014*). Te2.2 captured the most posterior portion of the larger tonotopic gradient that, consistently with previous reports, we identify as running in a direction orthogonal to Heschl's Gyrus (*Moerel et al., 2014*; *Besle et al., 2018*). The other cytoarchitectonic regions that overlapped with our functional acquisition field of view (Te2.1 and Te3) covered low-frequency preferring regions of the tonotopic map in the lateral portion of the Heschl's sulcus and the superior temporal gyrus. These results argue for the necessity of interpreting large-scale tonotopic maps, which by themselves do not allow defining the borders between superior temporal cortical areas (*Moerel et al., 2014*). A large tonotopic gradient unarguably runs in a posterior to anterior direction *Da Costa et al., 2011*; *Besle et al., 2018*. Equating this gradient with the gradient that identifies the primary auditory cortex results in a view in which the core lies orthogonal to Heschl's Gyrus (*Da Costa et al., 2011*; *Saenz and Langers, 2014*; *Besle et al., 2018*). On the other hand, the cytoarchitectonic areas -now restricted in size by better aligning macro-anatomy- suggest that the auditory core (Te1) runs along Heschl's Gyrus (i.e. the 'classical' view; *Dick et al., 2012*; *Moerel et al., 2014*). This view is strengthened by the combined interpretation of myelin and tonotopy (see *Figure 10* and results in *Dick et al., 2012*; *Moerel et al., 2014*) as well as other auditory cortical functional characteristics (e.g. frequency selectivity; *Moerel et al., 2014*).

Interesting differences exist between the surface projection of the cytoarchitectonic areas compared to a recent parcellation of the temporal lobe derived solely from in vivo imaging (*Glasser et al., 2016* - see *Figure 11*). Cytoarchitectonic areas Te1.1, Te1.0 and Te1.2 lie postero-medial to antero-lateral along the Heschl's Gyrus. The most lateral subdivision (Te1.2) has been suggested to be the human homologue of area RT in the monkey (and thus part of the auditory core) or part of the lateral belt (*Moerel et al., 2014*). In the multi-modal MRI parcellation, on the other hand, Heschl's Gyrus is divided in an area labeled as A1, corresponding to the most medial two thirds, and its most lateral portion, which is part of the area labeled as the medial belt. Outside of the Heschl's Gyrus, other differences between the in vivo and post-mortem atlas are visible. The lateral belt and parabelt areas as defined in the in vivo atlas occupy an area roughly corresponding to Te2, but the border between the areas labeled as belt and parabelt run approximately orthogonal to the border between Te2.1 and Te2.2. Te3, previously considered as an homologue of parabelt, corresponds to the areas labeled as A4 and A5 in the in vivo atlas. STS1 overlaps with the dorsal portion of superior temporal sulcus (STSda and STSdp in the in vivo atlas) and STS2 with the ventral portion of superior temporal sulcus for the most part. While the in vivo multi modal atlas has been derived from a large sample of participants (N = 210), these differences may be caused by an insufficient amount of information available in the in vivo data used for the parcellation of the superior temporal plane.

In conclusion, here we show that an alignment procedure tailored to the superior temporal cortex and driven by anatomical priors together with curvature improves inter-subject correspondence of cytoarchitectonic areas.Reducing macro-anatomical variability and improving cytoarchitectural correspondence may reduce the inter-subject variability of (anatomical and functional) characteristics

probed in vivo, resulting in a more accurate definition of putative cortical (temporal) areas. These results go beyond the test case we provide here (the analysis of 7 Tesla high-resolution data) and can benefit the auditory neuroscience community as they justify the use of a landmark based alignment approach when considering the temporal cortex. In particular, we expect a landmark approach to benefit studies that rely on inter individual alignment for statistical analysis (as also already demonstrated by *Kang et al., 2004*; *Desai et al., 2005*). In addition, the improved anatomical cytoarchitectonic atlas is a resource for all investigations (at both low and high fields) that rely on the definition of (cytoarchitectonically defined) regions of interest in their analyses. Finally, having validated a landmark-based alignment justifies its use in investigations of the anatomical and functional characteristics of auditory cortical areas using in vivo MRI. While we demonstrate its effectiveness in the temporal cortex, this approach is easily extendable to other cortical areas in which macro-anatomical inter subject variability is not easily accounted for by standard surface registration methods. Future studies should evaluate if this procedure, apart from being more accurate, is equally accurate for all known macro-anatomical variations of the morphology of the Heschl's Gyrus.

## Materials and methods

### Postmortem data

We used the cytoarchitectonically labeled temporal cortical areas of the 10 brains used in *Morosan et al., 2001*; *Zachlod et al., 2020*. The labeled areas were Te1.0, Te1.1, Te1.2, Te2.1, Te2.2 (*Morosan et al., 2001*), Te3 (*Morosan et al., 2005*), STS1, STS2 (*Zachlod et al., 2020*). All brains were linearly registered to Colin27 space (*Evans et al., 2012*) at 1 mm isotropic resolution, which was the starting point for all further analyses.

### Cortical segmentation

In order to perform CBA, the white matter - gray matter boundary was segmented in all 10 postmortem brains. The anatomical image quality was insufficient to employ fully automatic segmentation methods. To mitigate this issue, we employed a spatial filter that was applied to an upsampled version of the data (to 0.5 mm isotropic). This spatial filter was tailored to exploit the structure tensor field derived from the images. Our implementation of this procedure -that mostly follows *Weickert, 1998*; *Mirebeau et al., 2015*- included the following steps:

1. Smoothing the image for spatial regularization

$$\hat{v} = K_\sigma * v. \tag{1}$$

where * indicates convolution and $K$ is a Gaussian kernel with standard deviation defined by σ. Here we have opted for $\sigma = 1$.

2. Computing the gradients of the image to obtain a vector field (we have used central differences)

$$\mathrm{gradient}(\hat{v}) = \vec{v}. \tag{2}$$

3. Generating a structure tensor field by using the self outer product:

$$S = \vec{v} \cdot \vec{v}^{\mathrm{T}}. \tag{3}$$

4. Decomposing (using eigen decomposition) the structure tensor field:

$$\mathrm{eig}(S) \rightarrow \vec{e}_1, \vec{e}_2, \vec{e}_3 \,(\text{eigen vectors}) \,\text{and}\, \lambda_1, \lambda_2, \lambda_3 \,(\text{eigen values}). \tag{4}$$

Note that the eigen vectors are sorted according to eigen values $\lambda_1 > \lambda_2 > \lambda_3$.
5. Using eigen values to derive a vector field:

$$\begin{aligned}
\text{intensity} &= \lambda_1 + \lambda_2 + \lambda_3, \\
\text{range} &= (\lambda_1 - \lambda_3)/\text{intensity}, \\
W &= |(|\text{range} - 0.5| + 0.5) - \text{intensity}|.
\end{aligned} \tag{5}$$

Here we wanted to enhance prolate ellipsoid tensors (also called surfels, surface elements,

$\lambda_1 > \lambda_2 \approx \lambda_3$) more than isotropic structure tensors ($\lambda_1 \approx \lambda_2 \approx \lambda_3$) and oblate ellipsoid tensor (also called curvels, curve elements like tubes, $\lambda_1 \approx \lambda_2 > \lambda_3$).

6. Generating a diffusion tensor field from weighted eigen vectors:

$$(w \cdot e) = \mathrm{D}. \tag{6}$$

7. Smoothing the diffusion tensor field.

$$\hat{\mathrm{D}} = K_\rho * \mathrm{D} \tag{7}$$

where * indicates convolution and *K* is a 3D Gaussian kernel with σ standard deviation. Here, we have used $\rho = 1$. A higher value would enhance features at a larger spatial scale.

8. Computing a vector field (the flux field) using the diffusion tensor field and eigen vectors ($\mathrm{D}_i$ is a tensor $3 \times 3$; $\vec{v}_i$ is a vector $1 \times 3$):

$$\vec{f} = \hat{\mathrm{D}} \cdot \vec{v}. \tag{8}$$

9. Updating the image ($\vec{f}_i$ is a vector; $v_i$ is a scalar):

$$v_{\mathrm{new}} = v + \mathrm{divergence}(\vec{f}). \tag{9}$$

10. Repeating all steps until the desired number of iterations is reached (each iteration diffuses the image more and the diffusion is non-linear and anisotropic).

For segmenting the postmortem data, here we iterated this process 40 times. This number of iterations was visually judged as sufficient to enhance the boundary between white matter and gray matter as well as distinguishing the two banks of sulci by rendering them sharper (see *Figure 12*). Our implementation is available within the Segmentator package version 1.5.3 (*Gulban and Schneider, 2019*).

## Cortical surface reconstruction

After filtering the images, we generated an initial white matter segmentation using intensity-gradient magnitude joint 2D histograms (*Gulban et al., 2018b*). This initial segmentation was corrected in two stages. First, manual corrections were performed by O.F.G using both enhanced and un-enhanced anatomical images (around 8 hr of manual work per brain). Second, after splitting left and right hemispheres, we generated surfaces as triangular meshes using the marching cubes method (as implemented in BrainVoyager 21.4, *Goebel, 2012*) and decimating the total amount of vertices to 200,000 (with approximately equal edge lengths). The surfaces were visually checked for bridges and holes and problematic areas were corrected until the Euler characteristic of each surface became 2 (i.e. topologically identical to a sphere). *Figure 13* shows the morphological variation across the postmortem brains on the superior temporal cortex.

## Cortical surface alignment

The prepared surfaces were inflated to an approximate sphere and mapped onto a high-density spherical mesh (163842 vertices). Prior to CBA, the meshes were aligned using a spherical rigid body method to minimize curvature differences across subjects (see *Figure 1* left column). CBA was performed in two different ways. First, we non linearly registered the surfaces of each hemisphere across brains using standard cortex based alignment (i.e. minimizing curvature differences across individuals in a coarse to fine manner [*Frost and Goebel, 2012*]). Second, to tailor the alignment to the superior temporal cortices (left and right separately), we delineated four macro-anatomical landmarks: 1) the anterior Heschl's Gyrus; 2) the superior temporal gyrus; 3) the superior temporal sulcus and 3) the middle temporal gyrus (see *Figure 13* and *Figure 14*). Each of the landmarks was defined on an inflated representation of the hemisphere by identifying major landmark points and drawing a line segment between them that took into account the local curvature (i.e. minimizing the geodesic distance between the landmarks identifying each line). In particular, we defined the anterior Heschl's Gyrus by drawing a line segment between the most medial tip of the gyrus to the most lateral point where it merges with the superior temporal gyrus. We have drawn superior temporal gyrus, superior temporal sulcus, and medial temporal gyrus as three (mostly) parallel lines along posterior to

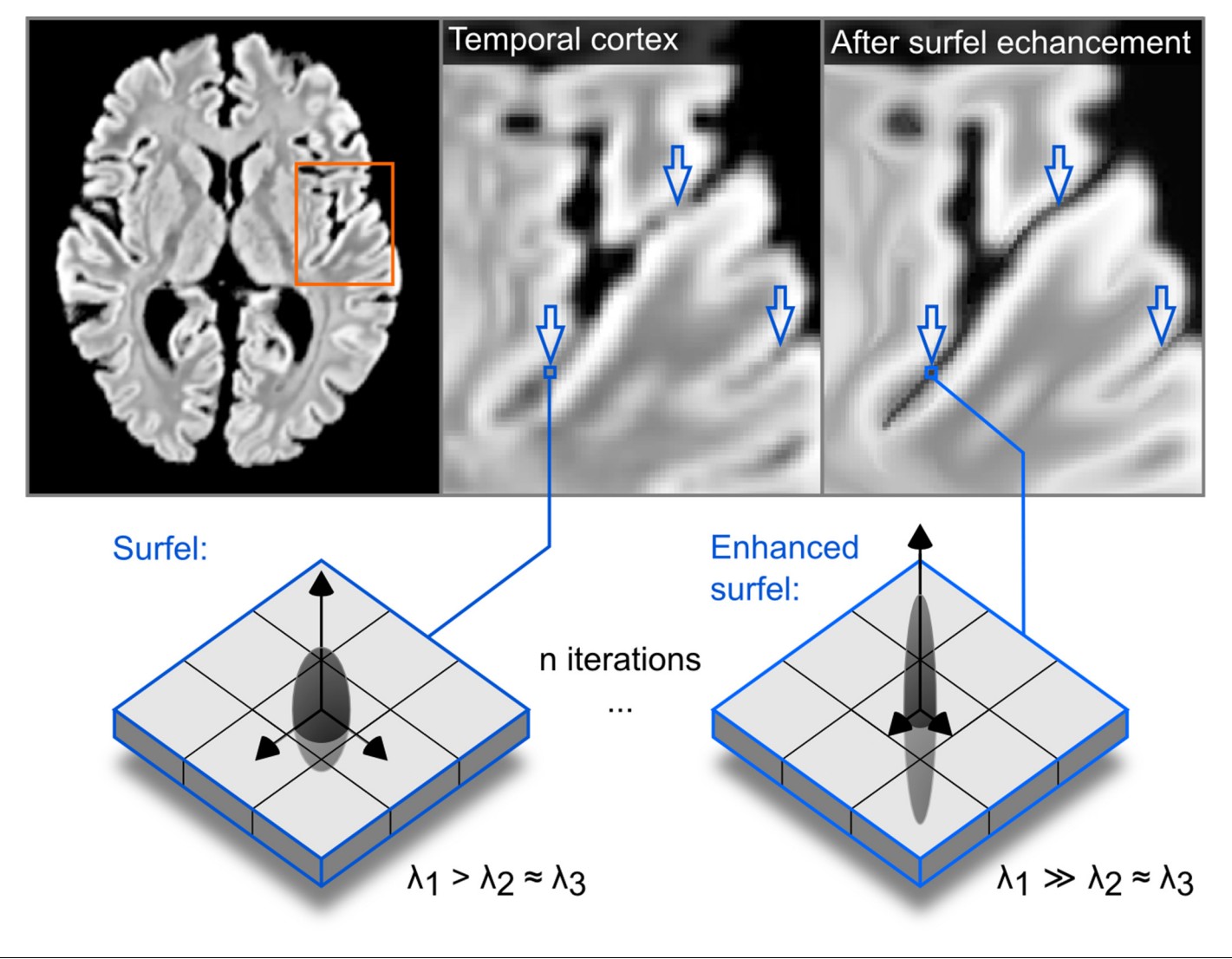

**Figure 12.** The effect of the structure enhancing filter shown on a transversal slice.  Blue arrows point to locations where local contrast is sharpened.

anterior direction. A tutorial describing how these anatomical priors are drawn together with several related resources is available at: https://github.com/ofgulban/cortical-auditory-atlas. These landmarks were used as additional information to determine the cost that is minimized during curvature based non-linear alignment in our tailored approach (i.e. CBA+). As for the standard surface alignment, CBA+ was performed across four spatial scales (from very smooth to slightly smooth curvature maps). Note that the procedure smooths both the global curvature map and the local landmark definitions (i.e. creates smooth maps from the landmarks). Apart from improving overall alignment (*Frost and Goebel, 2012*; *Tardif et al., 2015*), this procedure improves the robustness of the approach to the variability in landmark definition across users. Both CBA and CBA+ were performed using dynamic group averaging. The surface alignment yielded a mapping between each individual to a group average brain, each consisting of the same number of vertices.

To evaluate the effect of alignment, we computed the overlap across individuals for each of the cytoarchitectonic areas. We compared our tailored alignment procedure to the original volumetric Colin27 alignment (*Evans et al., 2012*), spherical rigid body alignment, and non-linear standard cortex based alignment (see *Figure 14*).

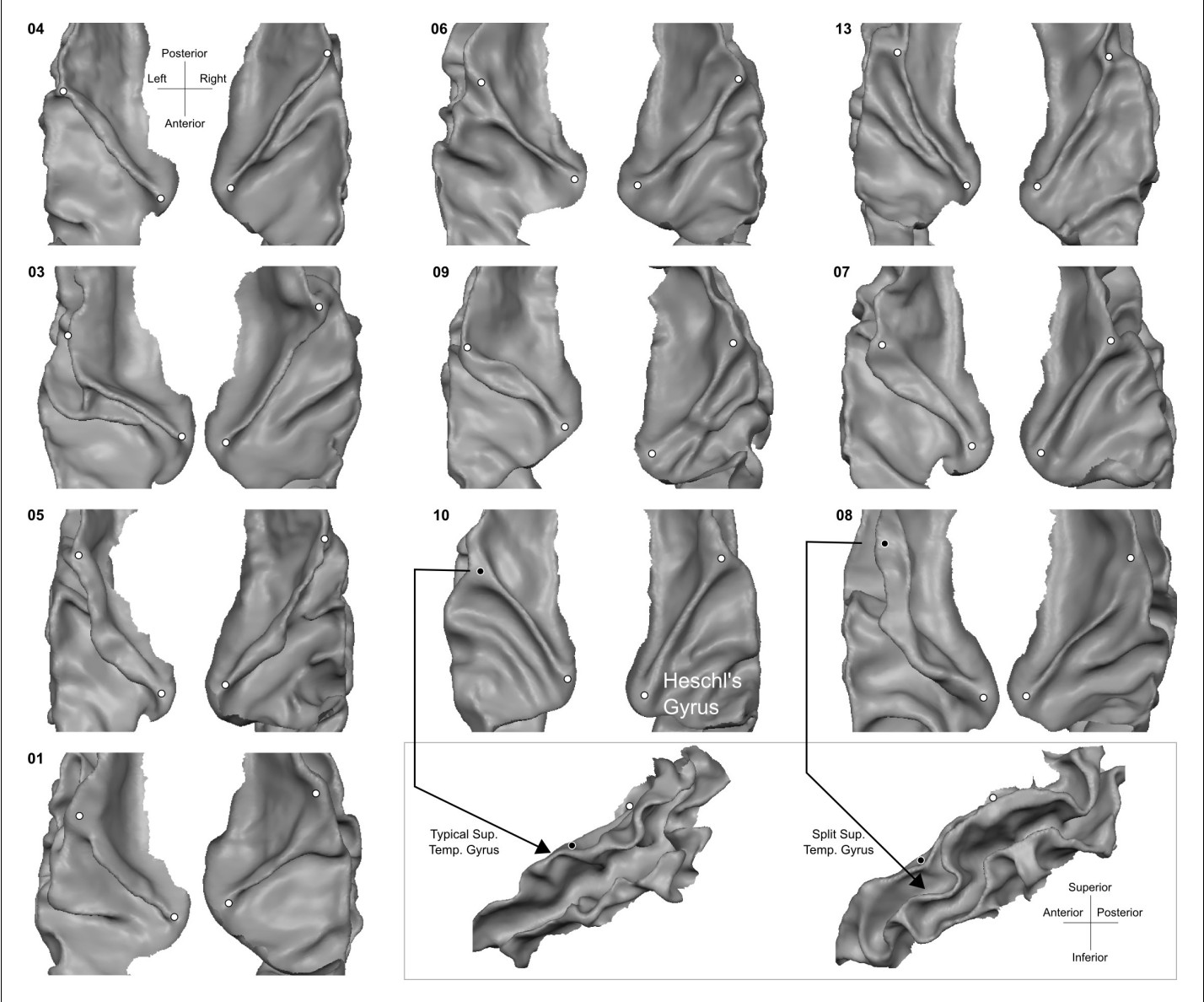

**Figure 13.** Individual superior temporal cortex white-gray matter boundary reconstructions. Anterior Heschl's Gyrus is indicated as the gyrus between white dots. The bottom right side shows the rare occurrence of a split superior temporal gyrus (*Heschl, 1878*) in contrast to a typical superior temporal gyrus from the side view.

## In vivo data

### MRI acquisition

We have used the dataset (This dataset is available at: https://openneuro.org/datasets/ds001942/versions/1.2.0) described in *Sitek et al., 2019*. This dataset includes: (I) $T_1$ weighted ($T_1$w), proton density weighted (PDw) and $T_2^*$ weighted ($T_2^*$w) anatomical images collected (using a modified MPRAGE sequence) at a resolution of 0.7 mm isotropic (whole brain); (II) functional images at collected at a resolution of 1.1 mm isotropic (partial coverage, coronal-oblique slab, multi-band factor = 2; GRAPPA = 3) in response to the presentation of natural sounds (168 natural sounds; 24 runs divided in four cross validation splits of 18 training and 6 testing runs each (126 training sounds and 42 testing sounds per split).

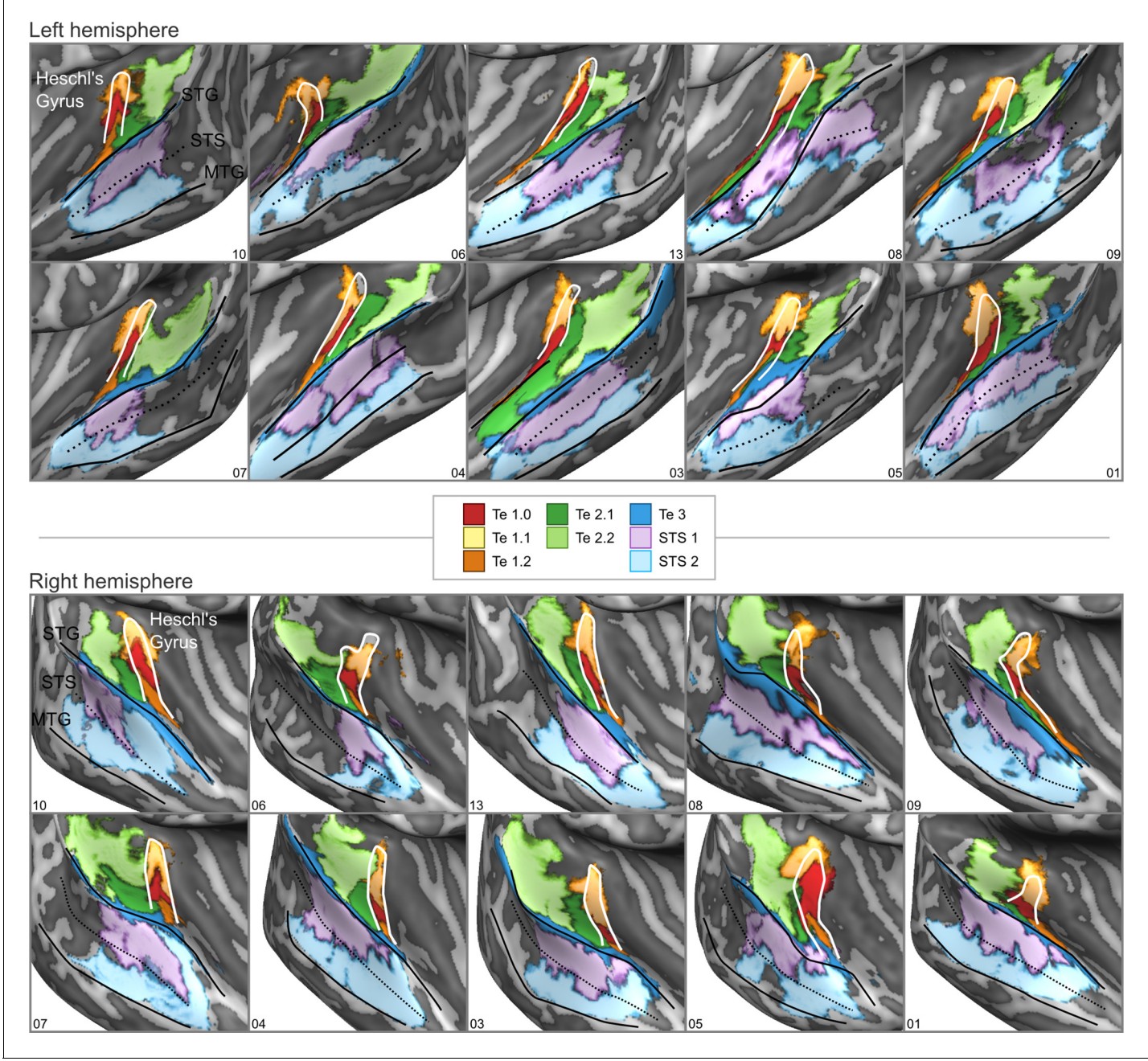

**Figure 14.** cytoarchitectonic areas of *Morosan et al., 2001*; *Morosan et al., 2005*; *Zachlod et al., 2020* sampled on the inflated cortical surfaces for each individual brain in the postmortem dataset. Anterior Heschl's Gyrus, superior temporal gyrus (STG), superior temporal sulcus (STS), and middle temporal gyrus (MTG) are indicated as line drawings.

## Cortical segmentation and alignment

Segmentations of both the white matter - gray matter interface and outer gray matter (also called gray matter - cerebrospinal fluid interface) were done following BrainVoyager 2.8.4's advanced segmentation pipeline (*Kemper et al., 2018*) and using the automatic bridge removal tool (*Kriegeskorte and Goebel, 2001*). Manual corrections were done in ITK-SNAP (*Yushkevich et al., 2006*). All follow-up analyses were performed by sampling (anatomical and functional) data onto the middle gray matter surfaces (defined using the equidistant methods [*Waehnert et al., 2014*; *Kemper et al., 2018*] by the combination of inner and outer gray matter surfaces). This allowed us

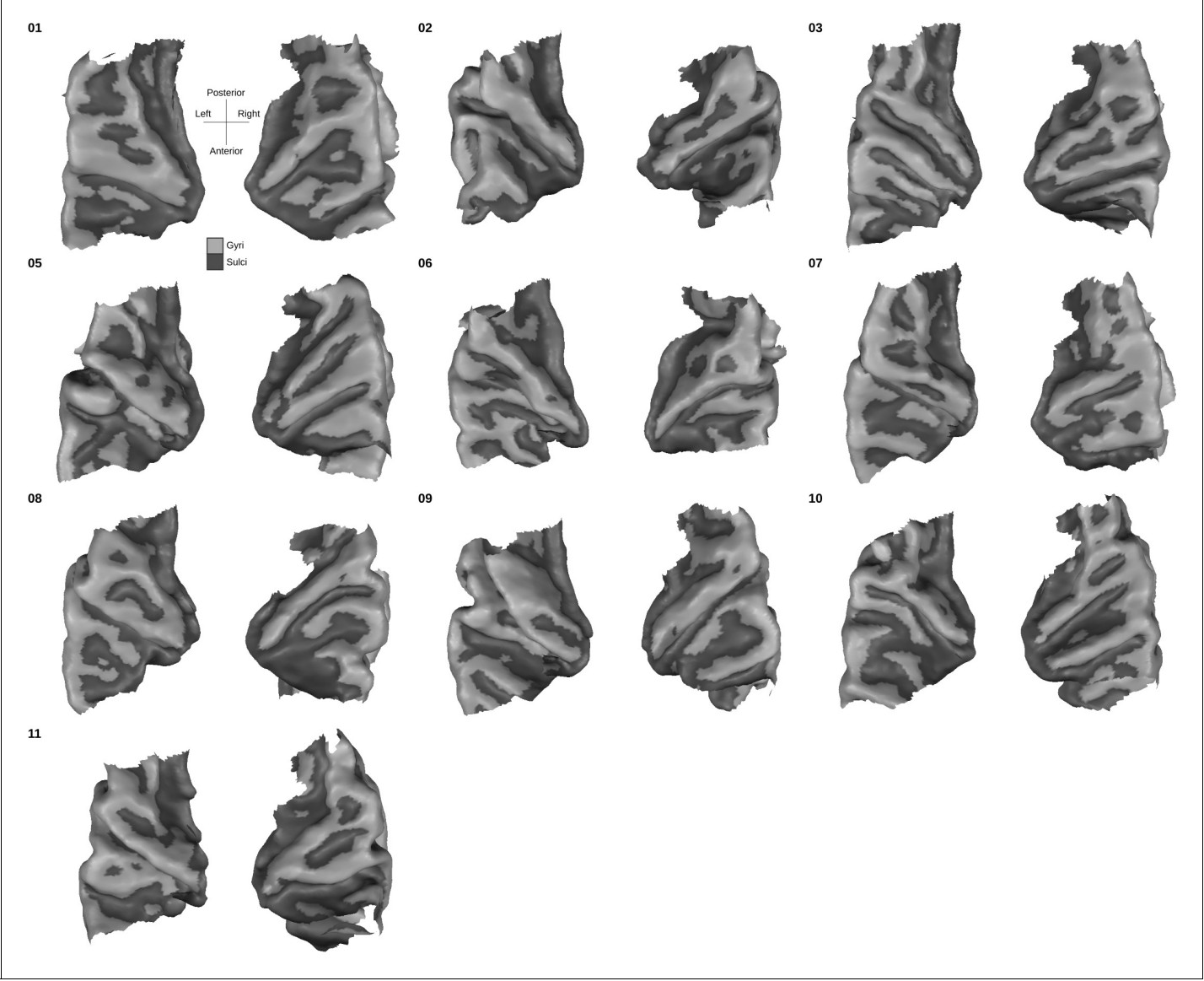

**Figure 15.** Individual superior temporal cortex middle-gray matter surface reconstructions from a bird's eye (top-down) view. Dark gray colored indicate sulci and light gray indicates gyri.

to minimize partial voluming with white matter, cerebrospinal fluid or superficial vessels. These surfaces can be seen for each individual in *Figure 15*.

The middle gray matter surfaces of all individuals were aligned using the procedure tailored to the superior temporal plane described above (CBA+). The resulting group average mesh from the in vivo dataset was aligned to the average postmortem mesh following the same procedure. This allowed us to overlay probabilistic cytoarchitectonic areas onto the in vivo group average cortical surfaces and sample functional and anatomical data within each area.

## Myelination maps

The processing steps followed to create myelination maps were similar to *De Martino et al., 2015*. $T_1$w images were divided by $T_2^*$w ($T_1$w/$T_2^*$w) and the resulting division image was masked by the cortical gray matter segmentation. A histogram-based adaptive percentile threshold (based on iterative deceleration of percentile differences) on the $T_1$w/$T_2^*$w image was used to discard voxels with

extreme intensities corresponding to vessels and regions in which the $T_2^*$w data were of insufficient quality. Maps were rescaled to range between 0 and 100. This step was necessary to match intensity ranges across subjects since we did not have quantitative measures. Values in the middle gray matter of the rescaled maps were sampled onto the surface mesh.

### Tonotopy maps

The functional data were preprocessed using BrainvoyagerQX v2.8.4 (*Goebel, 2012*). Slice-scan-time correction, motion correction, temporal high-pass filtering (GLM-Fourier, six sines/cosines) and temporal smoothing (Gaussian, kernel width of two acquisition volumes [i.e. 5.2 s]) were applied. Default options in BrainvoyagerQX v2.8.4 were used aside from the explicitly stated values. The functional images were then distortion corrected using the opposite phase encoding direction images using FSL-TOPUP (*Andersson et al., 2003*) as implemented in *Smith et al., 2004*. The conversion between Brainvoyager file types to NIfTI, which was required to perform distortion correction, was done using Neuroelf version 1.1 (release candidate 2) (http://neuroelf.net/ in Matlab version 2016a).

After pre-processing, functional images were transformed to Talairach space using BrainvoyagerQX v2.8.4 at a resolution of 1 mm isotropic. We estimated the voxels' responses to each natural sound in a two-step procedure (*Moerel et al., 2013*; *Santoro et al., 2014*). First, the hemodynamic response function (HRF) best characterizing the response of each voxel was obtained using a deconvolution GLM (with nine stick predictors together with the noise regressors) on the training data (a subset of the functional runs). Second, the response to each natural sound (in training and test set runs separately per cross validation) was estimated using a GLM analysis and the optimized HRF of each voxel. In addition to the predictors representing the experimental conditions (i.e. the individual stimuli), the analysis included noise regressors obtained using GLM-denoise (*Kay et al., 2013*). Note that the number of noise components and their spatial maps (allowing to derive the temporal regressors) where estimated on the training data only (i.e. separately per each cross-validation).

To estimate the voxels' preference for the acoustic content (i.e. sound frequencies), we fitted (using Ridge Regression) the spectral sound representation obtained by passing the sounds through a cochlear filter model (128 logarithmically spaced filters, see *Chi et al., 2005*; *Moerel et al., 2013*) to the voxels' responses (i.e. linearized encoding approach [*Kay et al., 2008*]). The frequency associated with the largest linear weight after fitting defined the preference of each voxel (see *Moerel et al., 2012* for more details on the procedure). Tonotopic maps were obtained by color coding (red to blue) the frequency preference (low to high) at each voxel.

## Acknowledgements

We thank Peer Herholz, Agustin Lage-Castellanos, and Fred Dick for their comments and advice at different stages of this project. The authors OFG and FDM were supported by NWO VIDI grant 864-13-012, and MM was supported by NWO VENI grant 451-15-012. This project has received funding from the European Union's Horizon 2020 Framework Programme for Research and Innovation under the Specific Grant Agreement No. 785907 (Human Brain Project SGA2).

## Additional information

### Funding

| Funder | Grant reference number | Author |
| --- | --- | --- |
| Nederlandse Organisatie voor Wetenschappelijk Onderzoek | VIDI 864-13-012 | Omer Faruk Gulban Federico de Martino |
| Nederlandse Organisatie voor Wetenschappelijk Onderzoek | VENI 451-15-012 | Michelle Moerel |
| Human Brain Project | 785907 (SGA2) | Rainer Goebel Katrin Amunts |

The funders had no role in study design, data collection and interpretation, or the decision to submit the work for publication.

## Author contributions
Omer Faruk Gulban, Conceptualization, Resources, Data curation, Software, Formal analysis, Validation, Investigation, Visualization, Methodology, Writing - original draft, Project administration, Writing - review and editing; Rainer Goebel, Resources, Data curation, Software, Methodology, Writing - review and editing; Michelle Moerel, Writing - review and editing; Daniel Zachlod, Resources; Hartmut Mohlberg, Resources, Data curation; Katrin Amunts, Resources, Data curation, Funding acquisition, Writing - review and editing; Federico de Martino, Conceptualization, Resources, Software, Formal analysis, Supervision, Funding acquisition, Validation, Investigation, Visualization, Methodology, Writing - original draft, Project administration, Writing - review and editing

## Author ORCIDs
Omer Faruk Gulban [iD] https://orcid.org/0000-0001-7761-3727
Katrin Amunts [iD] https://orcid.org/0000-0001-5828-0867

## Ethics
Human subjects: The experimental procedures were approved by the ethics committee of the Faculty for Psychology and Neuroscience at Maastricht University (reference number: ERCPN-167_09_05_2016), and were performed in accordance with the approved guidelines and the Declaration of Helsinki. Written informed consent was obtained for every participant before conducting the experiments. All partici- pants reported to have normal hearing, had no history of hearing disorder/ impairments or neurologi- cal disease.

## Decision letter and Author response
Decision letter https://doi.org/10.7554/eLife.56963.sa1
Author response https://doi.org/10.7554/eLife.56963.sa2

# Additional files

## Supplementary files
• Transparent reporting form

## Data availability
Post-mortem dataset is provided within Human Brain Project system: https://kg.ebrains.eu/search/instances/Dataset/ff71a4d1-ea14-4ed6-898e-b92d95b3c446. In vivo dataset is previously published in: https://openneuro.org/datasets/ds001942/versions/1.2.0.

The following previously published datasets were used:

| Author(s) | Year | Dataset title | Dataset URL | Database and Identifier |
|---|---|---|---|---|
| Gulban OF, Sitek KR, Ghosh SS, Moerel M, De Martino F | 2019 | Auditory localization with 7T fMRI | https://doi.org/10.18112/openneuro.ds001942.v1.2.0 | OpenNeuro, 10.18112/openneuro.ds001942.v1.2.0 |
| Gulban OF, Goebel R, Zachlod D, Mohlberg H, Amunts K, De Martino F | 2020 | Cytoarchitectonic areas of human auditory cortex on individual brain surfaces | https://doi.org/10.25493/CFBA-S36 | EBRAINS, 10.25493/CFBA-S36 |
| Gulban OF | 2020 | Cortical Auditory Atlas | https://doi.org/10.17605/OSF.IO/4MJPN | Open Science Framework, 10.17605/OSF.IO/4MJPN |

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
