## [Decision Letter]

**Acceptance summary:**

The work projects cytoarchitectonic areas after the well-established Munich scheme onto the superior temporal plane defined by 7T MRI using two types of coregistration based on curvature based alignment. The authors make the case that one type, based on priors tailored to temporal lobe surface landmarks, gives a better surface map of the cytoarchitectonic areas that is more consistent across subjects. The work is a Tools and Resources report, which explains the advance well and provides a clear practical guide for readers who wish to use it to interpret functional imaging data from group or group-case-studies data.

**Decision letter after peer review:**

Thank you for submitting your article "Improving a probabilistic cytoarchitectonic atlas of auditory cortex using a novel method for inter-individual alignment" for consideration by *eLife*. Your article has been reviewed by three peer reviewers, one of whom is a member of our Board of Reviewing Editors, and the evaluation has been overseen by Barbara Shinn-Cunningham as the Senior Editor. The following individual involved in review of your submission has agreed to reveal their identity: Christopher I Petkov (Reviewer #2).

The reviewers have discussed the reviews with one another and the Reviewing Editor has drafted this decision to help you prepare a revised submission.

The work projects cytoarchitechtonic areas after the well-established Munich scheme onto the superior temporal plane defined by 7T MRI using two types of coregistration based on curvature based alignment. The authors make the case that one type, based on priors tailored to temporal lobe surface landmarks, gives a better surface map of the cytoarchitechtonic areas that is more consistent across subjects. There will never be a perfect map because the cytoarchitechtonics do not rigidly correspond to anatomical landmarks between subjects but this technique is potentially useful in studies of individuals or groups for getting an idea of localisation of activity with respect to the cytoarchitechtonic areas. And it seems to cope well with the common HG duplication. The reviewers raised the following issue that need to be addressed.

Essential revisions:

1) Clarification of the general potential advance

• The authors need to make a better case that the complex superior alignment process is 'worth it' for gaining insight into the auditory system in functional auditory studies, as opposed to informing debate about parcellation of the human auditory cortex. This is required for publication as a Tools and Resources article.

• As 7T studies have small sample sizes and often do analyze individuals' data separately, the need for and benefits of making a precise alignment between subjects as opposed to individual-level analysis needs to be clarified. In the Discussion, the authors note that the lack of a full tonotopic gradient in Te1.0 "may be the result of excessive smoothing caused by inter-subject averaging."

• The paper does not cite earlier studies that have used landmark based registration to improve human auditory cortex mapping (e.g. Kang et al., 2004). A more careful discussion of how the approach has moved on with CBA+ is warranted.

2) Clarification and quantification of the CBA+ advantage

• The paper is largely qualitative and the claim that the CBA+ is superior is not quantitatively demonstrated. This can be done in a number of different ways including some quantitative measure of the overlap maps or the distribution of overlap with the gold standard anatomical area probabilistic maps.

• Was any comparison made with techniques other than CBA?

• The authors make a point that individual subject maps may be important to show but do not actually show them.

• Since it seems the authors also have to select landmarks, it is worth knowing how reliable different raters might be in picking those landmarks and whether a specialist is needed to do so.

3) Practical implementation

• If the main contribution is the alignment technique itself, more information and perhaps resources about the technique's application could increase its impact. For example, how can other researchers go about using it? (For example, software access? How manual/involved is the use? The individual cyto-architectonic areas are made available, as are some of the other resources like the segmentation algorithm, but would a reader need to implement the CBA+ portion themselves?) Is the group-derived atlas somehow useful by itself to other researchers or was it more of a proof-of-concept? What are the applications and limitations? For example, can the parcellation be applied to 3T MRI data or is the 7T resolution necessary to benefit from CBA+?

• Because the manuscript has been submitted in the Tools and Resources category, the goal of which is to assist prospective users in deploying the technique within their own work, more could be done to guide the reader in the application of the CBA+ technique (subsection “Cortical surface alignment”). From the description, a new user might struggle to specify macro-anatomical landmarks or where to start running that portion of the analysis, which seems to be the critical advance.

• Software/scripts/tutorial steps to implement the technique on a sample dataset would be helpful (e.g. Stropahl et al., 2018).

---

## [Author Response]

Essential revisions:1) Clarification of the general potential advance• The authors need to make a better case that the complex superior alignment process is 'worth it' for gaining insight into the auditory system in functional auditory studies, as opposed to informing debate about parcellation of the human auditory cortex. This is required for publication as a tools and method piece.

We thank the reviewers for this comment. In the Discussion section of the revised manuscript we have included a section highlighting the relevance for functional auditory studies of both a more accurate cytoarchitectonic probabilistic parcellation and an alignment method that considers macro-anatomical landmarks in the temporal lobe (e.g. CBA+).

In particular, we argue that the large anatomical variability known to exist in the temporal lobe is detrimental to: (a) group studies that rely on inter-individual alignment; and (b) investigations in small samples (such as often conducted at high fields) that because of the reduced SNR rely on the identification of homologous cortical areas across individuals (allowing to average across voxels within a subject and across subjects). We state that:

“The superior temporal plane shows considerable macro-anatomical variability across individuals (Pfeifer, 1921, 1936; Von Economo and Horn, 1930; Rademacher et al., 1993; Zoellner et al., 2019). If unaccounted for, these large inter-individual differences are detrimental to functional (and anatomical) in vivo investigations of the temporal lobe as they limit the efficacy of alignment procedures that are used in group studies. […] As a consequence, investigators often rely on available parcellations (see, e.g., the procedure followed by Dick et al., 2012) which result from post-mortem (or in vivo) investigations in a population and as a consequence rely on the quality of the inter-subject alignment.”

Thus, a more accurate inter-subject alignment procedure validated on cytoarchitectonic data provides several benefits for the auditory neuroimaging community:

“Our results have two potential benefits for the auditory neuroscience community. […] This justifies the use of these approaches (the one we propose here or others that have been proposed before Kang et al., 2004; Desai et al., 2005) when aligning individual temporal cortices with each other (or to a template).”

• As 7T studies have small sample sizes and often do analyze individuals' data separately, the need for and benefits of making a precise alignment between subjects as opposed to individual-level analysis needs to be clarified. In the Discussion, the authors note that the lack of a full tonotopic gradient in Te1.0 "may be the result of excessive smoothing caused by inter-subject averaging."

We thank the reviewer for this comment. Although many early 7T studies focused on small sample sizes (or even single individuals), the limited SNR of laminar studies is pushing the community to collect data from increasingly larger samples (see e.g. Lawrence et al., 2019 with a sample of N=26 participants). In the revised Discussion we argue that the main advantage of our procedure for high field studies rests on the ability to more accurately locate cortical areas across individual brains. To overcome the limited SNR of laminar studies, averaging is required within a subject (within an area across voxels) and statistical analysis is performed across subjects (see e.g. Finn et al., 2019). This approach rests on the definition of homologous regions across individuals. While localizers can be used for some auditory areas, they do not exist for the vast majority of regions in the temporal lobe. For this reason, investigators may rely on available parcellations to define auditory cortical regions. CBA+ provides two significant advantages in this respect. First, the resulting cytoarchitectonic atlas is more accurate (as areas are characterised by an overall higher overlap) and thus provide a more accurate region definition for users of the atlas. In addition, the results obtained on the post-mortem sample justify the use of CBA+ when aligning an individual brain to the atlas. Minimizing the macro-anatomical differences of the in vivo sample to the atlas will guarantee a more accurate projection of the cytoarchitectonic areas to the individual anatomy. We discuss these benefits in the Discussion:

“Our results have two potential benefits for the auditory neuroscience community. […] This justifies the use of these approaches (the one we propose here or others that have been proposed before Kang et al., 2004; Desai et al., 2005) when aligning individual temporal cortices with each other (or to a template).”

The lack of a tonotopic gradient *within* Te1.0 in group maps can be caused by several factors. While CBA+ improves the correspondence across subjects (compared to other approaches), there could still be misalignment issues affecting this region. Borders could be inaccurate, or there could be misalignment of the gradient within a region. Finally, it may be that Te1.0 does not actually have a full gradient. To evaluate the quality of the alignment of the single individuals to the atlas, we now provide individual subject maps. These maps suggest that, at the individual level, a tonotopic gradient is present within Te1.0. A quantification of the information that can be leveraged by these maps in order to characterize these regions is, however, beyond the scope of this contribution. We discuss this in the revised text:

“While future studies are needed to quantify the information that can be leveraged from individual in vivo MRI data for characterizing the cytoarchitectonic areas, our preliminary results indicate that individual tonotopic [and myelin] maps (supplementary figures to Figure 10) provide more fine grained information (within smaller areas such as e.g. Te1.0) (Moerel et al., 2014).”

• The paper does not cite earlier studies that have used landmark based registration to improve human auditory cortex mapping (e.g. Kang et al., 2004). A more careful discussion of how the approach has moved on with CBA+ is warranted.

We thank the reviewer for noting this important omission. In the revised version of the manuscript we now cite the suggested reference and other relevant ones.

The procedure we use here (CBA+) is similar to the ones described in previous approaches, and we would like to summarize the main differences. First, CBA+ optimizes both the global pattern of curvature and the local landmarks (using weighting between the two) in a coarse to fine manner. The procedure described by Kang et al., 2004, is instead based on cutting the temporal lobe and thus does not consider global curvature. Second, CBA+ is a non-linear procedure (it uses the same approach of standard global curvature alignment but with a weighted information between global and local landmarks), while Kang et al., 2004, considers several linear procedures to align landmarks. Third, the landmarks described in Kang et al., 2004, are points (on the medial and lateral tip of the anterior Heschl’s gyrus as well as at the anterior and posterior end of the STG and on the middle MTG) while we define lines covering the same landmarks (a line on the crown of the anterior Heschl’s gyrus, one on the STG (anterior and posterior), one on the MTG and one on the STS). Some of these differences between CBA+ and the landmark approach described in Kang et al., 2004, are reconciled by Desai et al., 2004, that compares a local landmark approach based on lines (as in CBA+) with a global alignment approach in both cases using non-linear methods. Both references conclude that local approaches outperform volumetric alignment as well as global (spherical) alignment when considering small cortical regions in the temporal lobe. We discuss these differences in the revised manuscript Discussion:

“The necessity to use local landmarks to guide the alignment of temporal regions has been considered in previous research (Kang et al., 2004; Desai et al., 2005). […] Here we provided additional landmarks (the Heschl's Gyrus, the superior temporal gyrus/sulcus and middle temporal gyrus) to the CBA procedure and, differently from previous approaches (that considered only the local landmarks when aligning the temporal cortex), we optimized alignment of both local and global macro-anatomical features.”

While the algorithm we use is not entirely novel (the main difference with Desai et al., 2005 is the optimization of both local and global features), we would like to stress the relevance of the validation we perform using cytoarchitecture. The major shortcoming of previous studies is in fact that the validation of the alignment procedure rested solely on the improvement of the co-localization of macro-anatomical landmarks or the effect the alignment had on group functional studies. Nevertheless, the known variation of micro-anatomical features with respect to macro-anatomical landmarks in the temporal lobe (the location of PAC shifts with respect to the anterior HG in the case of incomplete duplications) questions whether alignment procedures do in fact reduce variance of co-localization of cortical areas across brains. As described in previous studies, the variability affecting group studies (when considering alignment algorithms) can be partitioned in: (1) the macro-anatomical variability; (2) the way that micro-anatomically defined cortical areas shift with respect to microanatomy; and (3) the way that functional characteristics are linked to micro-anatomy. We clarify our contribution in the revised Discussion:

“Tailoring the alignment of the temporal lobe to local landmarks is motivated by the expectation that minimizing macro-anatomy will consequently result in improved micro-anatomical alignment (i.e., the alignment of cortical areas in the temporal lobe) and that this in turn improves functional alignment (assuming that function and micro-anatomy co-localize). While previous studies have shown that approaches that use local landmarks result in improved group functional activation (Kang et al., 2004; Desai et al., 2005), here we addressed the much required validation of how well such an approach reduces the underlying micro-anatomical variability.”

We hope these changes clarify the connection/differences with previous landmark alignment methods used for the temporal cortex and the addition we provide by validating this approach based on cytoarchitecture data.

2) Clarification and quantification of the CBA + advantage• The paper is largely qualitative and the claim that the CBA+ is superior is not quantitatively demonstrated. This can be done in a number of different ways including some quantitative measure of the overlap maps or the distribution of overlap with the gold standard anatomical area probabilistic maps.

We thank the reviewer for the comments. We would like to clarify that, in the manuscript, we quantify the change in overlap between subjects for all the cytoarchitectonic areas using CBA+, CBA, rigid only alignment, and volumetric alignment. This is reported in Figures 2 – 7 in which we show the change between methods by means of the probabilistic maps (a brighter color indicates more overlap for that area across the ten postmortem brains) as well as the histograms of the overlap reported in Figures 8 and 9. In the revised manuscript, we have integrated these results by reporting dice coefficients for each of the cytoarchitectonic regions using the different approaches (Table 1 and 2). We performed this analysis using a leave-one-out procedure. For each hemisphere we considered the overlap with an atlas built from the remaining nine hemispheres. The Results section refers to the tables (subsection “Comparison between alignment methods”). Our results indicate that CBA+ improves the overlap of cytoarchitecture especially for regions along the HG or neighboring HG:

“Improving macro-anatomical correspondence resulted in improved overlap of the cyto-architectonic areas across subjects. […] The tailored approach (CBA+) resulted in increased overlap (also compared to standard CBA) in all areas but especially for those on Heschl's Gyrus or immediately adjacent to it (Te1.0, Te1.1, Te1.2 and Te2.1 – see Table 1 and Table 2).”

Here we do not perform analyses on the in vivo data we use and their potential improvement when using CBA+. The main reason for this is that the data we use (tonotopy and myelin) do not allow an easy definition of *areas*. Previous landmark approaches have looked at the improvement in the overlap of statistical maps stemming from a subtraction (e.g. phonemes > tones in Desai et al., 2004, or attend-auditory > attend-visual in Kang et al., 2004). Tonotopy, on the other hand, is built as a best frequency map (Formisano et al., 2003) and, as we discuss in our manuscript and others have done elsewhere, does not allow defining areas in the temporal lobe. Similarly, the overlap of maps reflecting intra cortical contrast (myelin) across subjects would critically depend on the threshold used for these maps.

Importantly, as stated in response to the previous comment, we believe the analysis of the changes in alignment of the cytoarchitectonic areas to be timely and a major contribution to this field. The ability of anatomical alignment methods to improve functional alignment rely on the crucial hypotheses that minimizing macro-anatomical variability reduces micro-anatomical variability and consequently functional variability (under the assumption that function and micro-anatomy are co-localized). Compared to previous research, here we provide a quantification of the reduction of micro-anatomical variability resulting from landmark based alignment approaches.

• Was any comparison made with techniques other than CBA?

The results on the post mortem data are always compared between CBA, CBA+, rigid only alignment (quantifying the improvement achieved when just sampling from volume to surfaces), and the standard volumetric alignment. We believe that comparing CBA+ (optimizing both global and local landmarks) to other local landmark approaches or nonlinear volumetric alignment is matter for future research. Importantly the publicly available dataset we provide (i.e. the segmentations and sampled cytoarchitectonic regions of the post mortem data) will facilitate these investigations. We now comment on this in the revised Discussion:

“In follow-up investigations, this publicly available data set could also be used to compare the performance of different landmark based approaches that use only local landmarks (Kang et al., 2004; Desai et al., 2005) as well as non-linear volumetric approaches to CBA+.”

• The authors make a point that individual subject maps may be important to show but do not actually show them.

We thank the reviewers for this comment. In the revised version of the manuscript we have included all individual subject maps (see new Figure 10—figure supplements 2-11) that resulted in the group average maps (Figure 10).

• Since it seems the authors also have to select landmarks, it is worth knowing how reliable different raters might be in picking those landmarks and whether a specialist is needed to do so.

We thank the reviewer for this comment. We have now provided a tutorial describing the use of the approach that includes a description of the definition of the landmarks (https://github.com/ofgulban/cortical-auditory-atlas).

With regard to the particular concern of the easiness of definition, we would like to clarify a few things that we have also now added to the manuscript. We define the landmarks by drawing a line along major macro-anatomical features (HG, STG, STS, MTG). These lines minimize geodesic distance (i.e. they are drawn by defining consecutive points and a line is drawn in order to account for curvature). This feature allows to improve inter-rater stability as it requires only the definition of intermediate points. Apart from describing the way we selected our anchoring points (e.g. the most medial tip of the anterior HG, etc.), the provided tutorial allows a clear visual interpretation of these landmarks. In the revised manuscript we describe this (and refer to the tutorial):

“Each of the landmarks was defined on an inflated representation of the hemisphere by identifying major landmark points and drawing a line segment between them that took into account the local curvature (i.e. minimizing the geodesic distance between the landmarks identifying each line). […] A tutorial describing how these anatomical priors are drawn together with several related resources is available at: https://github.com/ofgulban/cortical-auditory-atlas”.

In addition, we would like to note that the algorithm (CBA+) uses a coarse to fine alignment approach and in doing so it actually generates maps from the line landmarks at various levels of smoothing. This is relevant as it allows to improve the alignment (starting from smoother landmarks allows to account better for large displacements between subjects). At the same time, this approach introduces a certain level of regularization to the landmark drawing procedure. This methodological detail is now described as follows:

“Note that the procedure smooths both the global curvature map and the local landmark definitions (i.e. creates smooth maps from the landmarks). Apart from improving overall alignment (Frost and Goebel, 2012; Tardif et al., 2015), this procedure improves the robustness of the approach to the variability in landmark definition across users.”

3) Practical implementation• If the main contribution is the alignment technique itself, more information and perhaps resources about the technique's application could increase its impact. For example, how can other researchers go about using it? (For example, software access? How manual/involved is the use? The individual cyto-architectonic areas are made available, as are some of the other resources like the segmentation algorithm, but would a reader need to implement the CBA+ portion themselves?) Is the group-derived atlas somehow useful by itself to other researchers or was it more of a proof-of-concept? What are the applications and limitations? For example, can the parcellation be applied to 3T MRI data or is the 7T resolution necessary to benefit from CBA+?

We thank the reviewers for these comments. In the revised version of the manuscript we have included descriptions and text to clarify the technique’s application and impact. In particular we have:

a) Clarified the way we defined the landmarks, and provided a guide that should help users to perform the same analyses (available at: https://github.com/ofgulban/cortical-auditory-atlas).

b) By clarifying the difference between our approach and previous landmark based approaches, we hope we have also clarified that our results provide a useful validation of those approaches (i.e. we expect all landmark based approaches to result in an overall better alignment of cytoarchitectonic areas). We think this increases the scope of our results; it does not limit them to the use of the specific proprietary software (BrainVoyager). In the Discussion we state that:

“Second, we show that minimizing a particular set of macro-anatomical features results in a more accurate micro-anatomical alignment, and thus that using local landmarks for aligning temporal regions results in a better alignment of cytoarchitectonic areas. This justifies the use of these approaches (the one we propose here or others that have been proposed before (Kang et al., 2004; Desai et al., 2005)) when aligning individual temporal cortices with each other (or to a template).”

c) We believe that the group derived atlas is a useful resource for the wider community (not limited to 7T investigations). The resulting areas are smaller and characterised by a higher inter-individual overlap, and are thus a more accurate reference. In the revised Discussion we state that:

“Our results have two potential benefits for the auditory neuroscience community. First, by providing a more accurate (i.e. with improved inter subject alignment) probabilistic atlas of the cytoarchitectonic areas in the temporal lobe, we provide a valuable resource for those studies that rely on a parcellation scheme.”

d) Finally, we clarify that we do not consider these to be benefits only for high resolution functional studies. Previous landmark based methods have shown improvements also when considering functional studies at lower spatial resolutions.

“These results go beyond the test case we provide here (the analysis of 7 Tesla high resolution data) and can benefit the auditory neuroscience community as they justify the use of a landmark based alignment approach when considering the temporal cortex. […] Finally, having validated a landmark based alignment justifies its use in investigations of the anatomical and functional characteristics of auditory cortical areas using in vivo MRI.”

• Because the manuscript has been submitted in the "Tools and Resources" category, the goal of which is to assist prospective users in deploying the technique within their own work, more could be done to guide the reader in the application of the CBA+ technique (subsection “Cortical surface alignment”). From the description, a new user might struggle to specify macro-anatomical landmarks or where to start running that portion of the analysis, which seems to be the critical advance.

We thank the reviewers for their suggestion. We now provide a guide (available at: https://github.com/ofgulban/cortical-auditory-atlas) describing the approach that we believe will clarify its use. This platform is also intended to be a hub for giving directions for specific datasets used in this study and answering practical questions if/when they arise while being completely open to the public for future contributions.

• Software/scripts/tutorial steps to implement the technique on a sample dataset would be helpful (e.g. Stropahl et al., 2018).

We thank the reviewers for their suggestion. We believe the tutorial we provide (see previous point) can serve as a guide of the steps to follow (tutorial available at: https://github.com/ofgulban/cortical-auditory-atlas). Data conversion scripts are also made available in the same platform.